# Unlocking Cross-Modal Biosignal Synthesis: A Temporally-Aware VAE-Diffusion Model

Chenyang Xu [1]   Dezhen Wang [2]   Hao Wang [1]

## Abstract

Synthesizing authentic phonocardiograms (PCG) from ubiquitous electrocardiograms (ECG) is a critical task for accessible cardiac monitoring. Existing generative models, however, struggle to capture the heart's complex electromechanical coupling, failing to meet the dual requirements of temporal precision and physiological fidelity needed for clinically relevant waveform analysis. We introduce the Temporally-Aware VAE-Diffusion model, a synergistic hybrid architecture that resolves this trade-off. Our architecture enforces tight physiological coupling through an Enhanced Condition Fusion mechanism and explicitly models long-range cardiac dynamics via Temporal Attention Blocks. On the EPHNOGRAM benchmark, our model sets a new state-of-the-art, achieving a Pearson correlation of $0.810\pm0.008$, 95.95% S1 detection accuracy, and a precise 12.0 ms timing error, significantly outperforming leading diffusion and Transformer baselines. Crucially, our work provides a reproducible zero-shot transfer evaluation for ECG-to-PCG synthesis. Evaluated on the synchronized PhysioNet/CinC 2016 training-a/MITHSDB subset without target-domain training, our model preserves high waveform fidelity and clinically relevant timing structure under domain shift, including on pathological recordings. These results support cross-dataset robustness of the proposed synthesis framework, while downstream diagnostic validation remains an important direction for future work.

[1]Faculty of Cybersecurity Cryptology,Xidian University, Xi'an, China [2]School of Computer Science and Technology, Tongji University, Shanghai, China. Correspondence to: Hao Wang <haow@ieee.org>.

*Proceedings of the 43rd International Conference on Machine Learning*, Seoul, South Korea. PMLR 306, 2026. Copyright 2026 by the author(s).

## 1. Introduction

Cardiovascular disease remains the leading cause of mortality worldwide, motivating scalable and low-cost solutions for cardiac assessment. With the digital transformation of healthcare and rapid advances in AI (Ribeiro et al., 2020; Rajpurkar et al., 2022), remote monitoring and telemedicine have become increasingly feasible. Among cardiac signals, the electrocardiogram (ECG) captures the heart's electrical activity, while the phonocardiogram (PCG) records its mechanical and acoustic response. These modalities provide complementary diagnostic information (Goldberger et al., 2000; Kiyasseh et al., 2021), yet PCG acquisition still relies on specialized sensors and controlled recording conditions (Oliveira et al., 2021), limiting its availability in low-resource settings and consumer devices. Synthesizing PCG directly from ubiquitous ECG would therefore enable dual-modality cardiac assessment without additional hardware.

Achieving this synthesis is challenging because it requires modeling the heart's complex, patient-specific electromechanical coupling. The temporal relationship between ECG and PCG is shaped by heart-rate variability and pathology (Hannun et al., 2019; Attia et al., 2019), demanding both temporal precision and physiological fidelity. Existing methods fall broadly into two classes, each with inherent limitations. Regression-based models, such as Transformers, can preserve temporal alignment but tend to produce oversmoothed, averaged waveforms that lack the rich acoustic texture of real PCG. In contrast, standalone generative models, such as pure diffusion networks, can synthesize perceptually realistic audio but struggle with strict conditioning, often compromising physiological coupling and precise timing. This leads to a persistent trade-off between temporal coherence and perceptual quality.

We address this gap with a **Temporally-Aware VAE-Diffusion model**, a hybrid architecture that jointly targets high-fidelity synthesis and accurate ECG–PCG alignment. A Variational Autoencoder (VAE) first learns a structured latent space for PCG, providing a stable manifold for generation, while a conditional diffusion model operates in this latent space to produce detailed, realistic signals guided by the input ECG. To better capture electromechanical dynam-

ics, we introduce an Enhanced Condition Fusion block and a Temporal Attention mechanism, which explicitly model ECG-informed temporal dependencies. Through this design, our method achieves state-of-the-art performance on a standard benchmark and demonstrates strong cross-dataset zero-shot transfer on an unseen synchronized ECG–PCG subset, supporting its potential as a synthesis and data-augmentation tool while leaving diagnostic-task validation to future work.

## 2. Related Work

### 2.1. Conditional Generative Models in Healthcare

Conditional GANs (Chen et al., 2022; Zhang & Babaeizadeh, 2021), pioneered by Mirza & Osindero (2014), have been extensively adapted for medical applications. However, the instability of GAN training becomes particularly problematic in biosignal applications where mode collapse can eliminate critical pathological variations. Recent stabilization techniques by Karras et al. (2020) have improved the training dynamics, but application to time-series remains a challenge. Conditional VAEs offer more stable training but traditionally suffer from posterior collapse and blurred outputs. Overcoming these fidelity limitations, diffusion models (Ho et al., 2020; Karras et al., 2022; Tashiro et al., 2021) have emerged as powerful generative alternatives, achieving state-of-the-art results in image synthesis. Recent work has explored conditional diffusion models (Saharia et al., 2022; Li et al., 2023; Alcaraz & Strodthoff, 2023) and classifier-free guidance (Ho & Salimans, 2021), showing remarkable improvements in controllable generation. Kong et al. (2021) applied diffusion models to audio synthesis, demonstrating their potential for time-series data. Furthermore, Cardoso et al. (2023) introduced a diffusion-based ECG generator trained on healthy data that produces realistic signals and supports key clinical tasks. However, direct application to complex biosignals requires careful consideration of physiological constraints and temporal dependencies that are not present in general audio synthesis.

### 2.2. Temporal Modeling in Physiological Signals

Temporal modeling (Ding et al., 2024) is crucial for physiological signal processing due to the inherent time-varying nature of biological systems. Specifically for cardiac signals, temporal modeling must capture both periodic patterns and subtle variations. Traditional approaches relied on Hidden Markov Models and Kalman filters, but deep learning has enabled more sophisticated temporal representations. Hannun et al. (2019) demonstrated that deep learning could exceed the performance of cardiologists in the detection of arrhythmias using careful temporal modeling. Attia et al. (2019) showed that an AI-enabled ECG could identify patients with atrial fibrillation, relying heavily on temporal patterns invisible to human experts. Recent reviews such as Hong et al.

(2020) summarize opportunities and challenges for deep learning on ECG data. Transformer-based and generative architectures have further expanded the ability of biosignal models to capture complex long-range dependencies.

### 2.3. Cross-Modal Medical Signal Synthesis

Recent work has increasingly explored generative modeling for physiological time-series synthesis. On one hand, diffusion-based approaches such as DiffECG (Neifar et al., 2024) and RDDM (Shome et al., 2024) have shown strong capability in related biosignal synthesis and translation tasks, with other work exploring unconditional ECG generation via 2D representations (Adib et al., 2023). However, these approaches have not been systematically evaluated on ECG-to-PCG translation, which requires modeling a more complex and variable electromechanical coupling between electrical activation and mechanical heart sounds. In parallel, regression-based sequence-to-sequence models, notably the Transformer (Vaswani et al., 2017), effectively capture long-range temporal dependencies (Zerveas et al., 2021), but are fundamentally constrained by mean regression and tend to smooth the subtle, high-frequency acoustic details that are important for heart-sound morphology. This trade-off between generative fidelity and regression stability motivates a task-specific latent generative framework. Rather than proposing a new generic diffusion family, our work adapts latent diffusion to paired ECG-to-PCG synthesis through three design choices tailored to electromechanical coupling: hierarchical time-aligned ECG conditioning, a shared decoder manifold connecting reconstruction and generation, and joint fine-tuning that co-adapts the VAE manifold with the conditional diffusion prior.

**Relation to conditional time-series generation.** Recent conditional time-series generators such as Imagen-Time (Naiman et al., 2024a), SDformer (Chen et al., 2024), and KoVAE (Naiman et al., 2024b) broaden the design space beyond classical sequence-to-sequence regression and waveform diffusion. However, these methods primarily target generic temporal distribution modeling rather than paired, beat-level cardiovascular translation with explicit electromechanical coupling. Our contribution is therefore not a new generic diffusion family, but a task-specific synthesis framework that combines hierarchical time-aligned ECG conditioning, a shared decoder manifold, and joint latent/generative fine-tuning for ECG-to-PCG generation.

### 2.4. Evaluation Metrics and Timing-Oriented Validation

The evaluation of generative models in medical applications requires a rigorous balance between technical synthesis quality and downstream clinical relevance. While traditional distance metrics such as PSNR and SSIM are foundational, they often fail to capture waveform morphol-

ogy and event-level temporal structure that are important for clinically relevant analysis. To address this in generative modeling, (Heusel et al., 2017) introduced the Fréchet Inception Distance, and (Kynkäänniemi et al., 2019) proposed improved precision and recall distributions. For biosignals specifically, (Goldberger et al., 2000) early on emphasized the necessity of physiological plausibility.

Building upon this foundation, distribution-level metrics should be complemented with task-specific signal-structure evaluations. For ECG-to-PCG synthesis, this requires assessing the preservation of temporal dynamics and heart-sound event structure, rather than relying solely on generic reconstruction losses. Within the cardiovascular domain, PhysioNet challenges have helped standardize evaluation protocols for ECG and PCG analysis (Reyna et al., 2021; 2023). These community-driven protocols motivate our multi-faceted evaluation methodology, while downstream diagnostic validation remains necessary before any clinical use.

## 3. Methodology

We propose a **Temporally-Aware VAE-Diffusion** framework for cross-modal synthesis of PCG from ECG. The model follows a principled *representation–generation decoupling*: a VAE learns a temporally aligned and information-rich latent manifold for PCG, and a conditional diffusion model performs high-fidelity generation *on this manifold* rather than in raw waveform space. This design stabilizes learning and lets the denoiser allocate capacity to clinically relevant acoustic textures and precise event timing.

Two physiologically-motivated components drive temporal coherence between the electrical (ECG) and mechanical (PCG) signals: (i) **Enhanced Condition Fusion** for multiscale ECG guidance via cross-attention, and (ii) **Temporal Attention Blocks** for long-range dependencies (e.g., the electromechanical delay from R-peak to S1/S2).

### 3.1. Problem Formulation

Let $x^{\mathrm{ECG}} \in \mathbb{R}^L$ be an ECG segment and $x^{\mathrm{PCG}} \in \mathbb{R}^L$ the aligned PCG segment. We learn a conditional generative model $p_\theta(x^{\mathrm{PCG}} \mid x^{\mathrm{ECG}})$.

**Definition 3.1** (Latent conditional diffusion). We factorize generation into: (i) a VAE encoder $E_\phi$ mapping $x^{\mathrm{PCG}}$ to a structured latent map $\boldsymbol{z}_0 \in \mathbb{R}^{C' \times L'}$; (ii) a condition encoder $C_\omega$ mapping ECG to an aligned condition map $\boldsymbol{C} \in \mathbb{R}^{C_c \times L'}$; (iii) conditional diffusion to synthesize $\boldsymbol{z}_0$ from noise given $\boldsymbol{C}$; and (iv) a decoder $D_\psi$ reconstructing $\hat{x}^{\mathrm{PCG}}$ from $(\boldsymbol{z}_0, \boldsymbol{C})$.

### 3.2. Stage 1: VAE for Structured Latent Manifold Learning

**Motivation.**  Raw PCG waveforms are high-dimensional and noisy; directly learning a diffusion model in waveform space is empirically harder and tends to waste capacity on basic signal structure. We therefore learn a structured latent space that preserves temporal dynamics and local morphology, and let diffusion focus on synthesizing realistic textures on top of this scaffold.

**Encoder outputs.**  The encoder $E_\phi$ outputs a structured latent map $\boldsymbol{z}_0 \in \mathbb{R}^{C' \times L'}$. To mildly regularize the latent representation, we also parameterize a diagonal Gaussian posterior:

$$q_\phi(\boldsymbol{z} \mid x^{\mathrm{PCG}}) = \mathcal{N}\big(\boldsymbol{z};\, \boldsymbol{\mu}_\phi, \mathrm{diag}(\boldsymbol{\sigma}_\phi^2)\big). \qquad (1)$$

Sampling uses $\boldsymbol{z} = \boldsymbol{\mu}_\phi + \boldsymbol{\epsilon} \odot \boldsymbol{\sigma}_\phi, \boldsymbol{\epsilon} \sim \mathcal{N}(0, I)$.

**Dual-output latent design.**  The diffusion process uses the *map* $\boldsymbol{z}_0$ as the clean target at $t{=}0$, while the sampled vector $\boldsymbol{z}$ is used only for KL regularization during VAE training. This avoids an overly restrictive bottleneck while still preventing degenerate representations.

**Dimensionality convention.**  Throughout the paper, the structured diffusion target is the latent map $\boldsymbol{z}_0 \in \mathbb{R}^{128 \times 750}$ for a 12-second, 1 kHz segment. The stochastic posterior vector $\boldsymbol{z} \in \mathbb{R}^{32}$ is used only for KL regularization during VAE training. The 512-dimensional quantity appearing in the decoder and condition pathways denotes the projected working width after the FC or $1 \times 1$ convolutional input projection, not the diffusion latent dimensionality.

**Condition encoder.**  In parallel, $C_\omega$ maps the aligned ECG signal to a condition feature map $\boldsymbol{C} = C_\omega(x^{\mathrm{ECG}})$. Using a feature map (instead of a global vector) preserves temporal structure, enabling the denoiser to learn fine-grained alignment between ECG morphologies and heart sound events.

**Decoder with dual-path projection (training vs. inference).**  The decoder $D_\psi$ is conditioned on $\boldsymbol{C}$ and receives different latent inputs depending on the stage: during VAE training it reconstructs from the sampled vector $\boldsymbol{z}$, while during generation it reconstructs from the denoised map $\hat{\boldsymbol{z}}_0$ produced by diffusion. We implement this using two lightweight input projections (FC for $\boldsymbol{z}$, $1 \times 1$ Conv for $\boldsymbol{z}_0$) that merge into a weight-shared upsampling body, aligning the training and inference decoding pathway.

### 3.3. Stage 2: Conditional Latent Diffusion

We perform diffusion in latent space conditioned on $\boldsymbol{C}$.

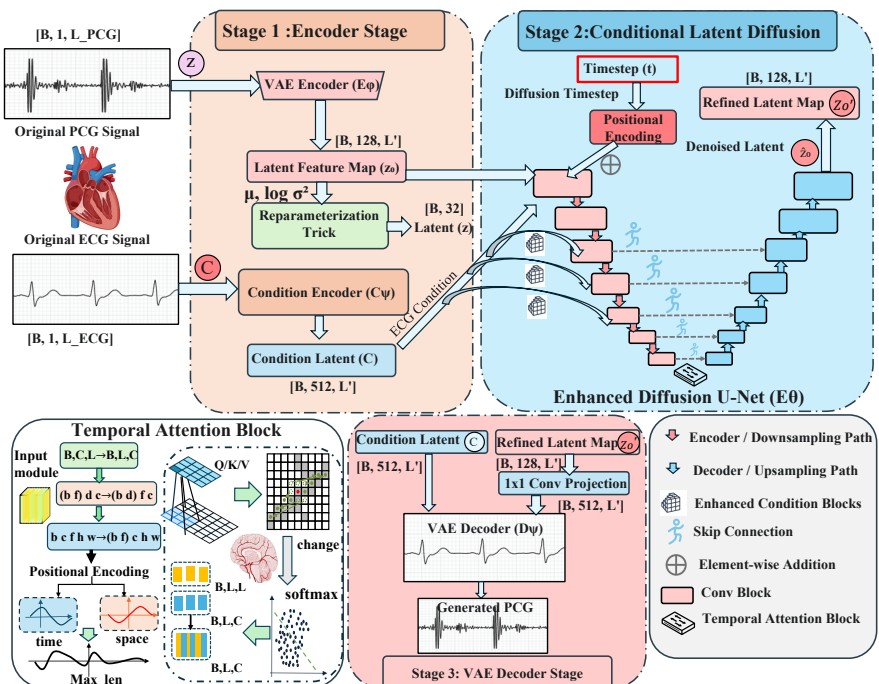

*Figure 1.* Overview of our Temporally-Aware VAE-Diffusion model. **Stage 1:** VAE encoder maps PCG to a structured latent map $\boldsymbol{z}_0$ (a derived low-dim vector is used only for regularization). **Stage 2:** a conditional diffusion U-Net denoises in the latent space guided by an ECG condition map $\boldsymbol{C}$. **Stage 3:** a weight-shared decoder reconstructs the PCG waveform.

### 3.3.1. FORWARD PROCESS

The forward process adds noise to $\boldsymbol{z}_0$:

$$q(\boldsymbol{z}_t \mid \boldsymbol{z}_0) = \mathcal{N}\big(\boldsymbol{z}_t; \sqrt{\bar{\alpha}_t}\boldsymbol{z}_0, (1 - \bar{\alpha}_t)I\big), \qquad (2)$$

where $\bar{\alpha}_t = \prod_{i=1}^{t}(1 - \beta_i)$. We use $T = 200$ steps with a linear schedule $\beta_1 = 10^{-4} \rightarrow \beta_T = 0.02$.

### 3.3.2. REVERSE PROCESS

We learn a conditional denoiser $\boldsymbol{\epsilon}_\theta(\boldsymbol{z}_t, t, \boldsymbol{C})$. One DDPM reverse step is:

$$\boldsymbol{z}_{t-1} = \frac{1}{\sqrt{\alpha_t}}\Big(\boldsymbol{z}_t - \gamma_t\, \boldsymbol{\epsilon}_\theta(\boldsymbol{z}_t, t, \boldsymbol{C})\Big) + \sigma_t\boldsymbol{\eta}, \qquad (3)$$

where $\boldsymbol{\eta} \sim \mathcal{N}(0, I)$ and

$$\gamma_t = \frac{\beta_t}{\sqrt{1 - \bar{\alpha}_t}}. \qquad (4)$$

At inference time, we start from $\boldsymbol{z}_T \sim \mathcal{N}(0, I)$ and iterate to obtain $\hat{\boldsymbol{z}}_0$.

### 3.4. Enhanced Diffusion U-Net with Temporally-Aware Conditioning

**Backbone.** Our denoiser is a 1D U-Net operating on latent sequences. We use base width 64 and multipliers $[1, 2, 4, 8]$, GroupNorm (8 groups), SiLU activations, and an MLP-projected sinusoidal timestep embedding.

**Temporal attention blocks.** To capture long-range dependencies across cardiac cycles, we apply temporal self-attention:

$$\boldsymbol{h}' = \boldsymbol{h} + \text{Attn}(\boldsymbol{h}), \qquad (5)$$

with

$$\text{Attn}(\boldsymbol{h}) = \text{softmax}\big((\boldsymbol{Q}\boldsymbol{K}^\top)/\sqrt{d}\big)\boldsymbol{V}. \qquad (6)$$

We set $\boldsymbol{Q} = \boldsymbol{h}\boldsymbol{W}^Q$, $\boldsymbol{K} = \boldsymbol{h}\boldsymbol{W}^K$, $\boldsymbol{V} = \boldsymbol{h}\boldsymbol{W}^V$ with 8 heads. These blocks are placed in the encoder to establish hierarchical temporal guidance early, which empirically improves S1/S2 timing.

### 3.4.1. ENHANCED CONDITION FUSION

A naive baseline concatenates ECG features with U-Net features. We instead propose **Enhanced Condition Fusion** to inject multi-scale ECG cues and learn dynamic alignment.

First, we enrich the condition via multi-scale convolutions:

$$\boldsymbol{C}_{\text{multi}} = \text{Concat}\big(\text{Conv}_3(\boldsymbol{C}), \text{Conv}_5(\boldsymbol{C}), \text{Conv}_7(\boldsymbol{C})\big). \qquad (7)$$

Then we fuse $\boldsymbol{C}_{\text{multi}}$ into the U-Net state $\boldsymbol{h}$ using cross-attention:

$$\text{Fuse}(\boldsymbol{h}, \boldsymbol{C}_{\text{multi}}) = \text{softmax}\big((\boldsymbol{Q}_h \boldsymbol{K}_C^\top)/\sqrt{d}\big)\boldsymbol{V}_C. \qquad (8)$$

We compute $\boldsymbol{Q}_h = \boldsymbol{h}\boldsymbol{W}^Q$ and $\boldsymbol{K}_C, \boldsymbol{V}_C = \boldsymbol{C}_{\text{multi}}\boldsymbol{W}^{K,V}$. For downsampled U-Net levels, $\boldsymbol{C}$ is temporally interpolated to match resolution. Unless otherwise stated, we inject

condition features only in the encoder path. We use this setting as the preferred empirical trade-off because it slightly improves timing and perceptual quality over symmetric injection under our five-seed evaluation, while avoiding stronger claims of statistical separation.

### 3.5. Stage 3: Waveform Reconstruction

After denoising, we decode the latent map:

$$\hat{x}^{\text{PCG}} = D_\psi(\hat{z}_0, C). \tag{9}$$

### 3.6. Training Objective

We optimize a composite objective

$$\mathcal{L} = \mathcal{L}_{\text{VAE}} + \lambda \mathcal{L}_{\text{diff}}, \tag{10}$$

with $\lambda = 1.0$. The VAE loss is

$$\mathcal{L}_{\text{VAE}} = \|x^{\text{PCG}} - \hat{x}^{\text{PCG}}_{\text{rec}}\|_2^2 + \beta \, D_{\text{KL}}\big(q_\phi(z \mid x^{\text{PCG}}) \,\|\, \mathcal{N}(0, I)\big), \tag{11}$$

where $\hat{x}^{\text{PCG}}_{\text{rec}} = D_\psi(z, C)$. The diffusion objective predicts the added noise:

$$\mathcal{L}_{\text{diff}} = \mathbb{E}_{t,\epsilon}\big\|\epsilon - \epsilon_\theta(z_t, t, C)\big\|_1. \tag{12}$$

**Implicit temporal coherence.** We intentionally avoid handcrafted temporal-alignment losses (e.g., enforcing explicit R-to-S1 constraints). Instead, we incorporate inductive biases through Enhanced Condition Fusion and Temporal Attention, allowing the model to learn electromechanical coherence directly from paired data. This design is empirically more robust and generalizes better across datasets and pathologies.

**Scope of the objective.** We do not claim that Eq. (10) is a single closed-form ELBO for the entire three-phase pipeline. Instead, the optimization should be interpreted stage-wise. Phase 1 optimizes a standard VAE reconstruction-plus-KL objective for learning an informative PCG latent manifold. Phase 2 optimizes a standard conditional denoising objective for latent diffusion with the VAE fixed. Phase 3 uses Eq. (10) as a surrogate joint fine-tuning objective that co-adapts the latent manifold, diffusion prior, and shared decoder. This distinction is important because the joint fine-tuning phase is designed for empirical alignment of the generation and reconstruction pathways rather than for preserving a closed-form variational bound.

### 3.7. Phased Optimization and Key Hyperparameters

We use a three-phase curriculum.

**Phase 1: VAE pre-training.** We train $(E_\phi, C_\omega, D_\psi)$ with AdamW (lr $1 \times 10^{-4}$, batch size 32, cosine annealing). The

KL weight is annealed from 0 to $\beta_{\text{final}} = 10^{-6}$ to preserve high-frequency details while maintaining mild regularization.

**Phase 2: latent diffusion training.** We freeze VAE parameters and train $\epsilon_\theta$ using Eq. (12) with $T = 200$.

**Phase 3: joint fine-tuning.** We unfreeze all parameters and optimize Eq. (10) with lr $1 \times 10^{-5}$, aligning the diffusion-generated latent distribution with the shared decoder.

### 3.8. Inference and Sampling-Speed Trade-off

We report results using DDPM sampling with $T = 200$ steps. For low-latency settings, we also use deterministic DDIM sampling with fewer steps while keeping weights fixed, yielding near-linear speedups with graceful quality degradation.

### 3.9. Design Choices and Empirical Justification

We validate key design choices via controlled ablations: (i) latent diffusion vs. waveform diffusion, (ii) cross-attention fusion vs. channel-wise concatenation, (iii) encoder-only vs. alternative conditioning injection, and (iv) low-$\beta$ KL regularization for preserving fine-grained acoustic details.

## 4. Experimental Evaluation

We validate our model through rigorous evaluation of **in-distribution performance**, **zero-shot generalization**, and **component-wise ablation**.

### 4.1. Experimental Setup

**Datasets.** We use **EPHNOGRAM** (Kazemnejad et al., 2021) as the source dataset for training, validation, and in-distribution testing. Data are split at the subject level into 70%/15%/15% train/validation/test partitions. For zero-shot transfer, we use the synchronized **training-a/MITHSDB subset** of the **PhysioNet/CinC Challenge 2016** dataset (Clifford et al., 2016), which contains paired single-lead ECG and PCG recordings. This target subset is used only for evaluation; no target-domain samples are used for training or model selection.

**Zero-Shot Protocol.** The ECG condition is the synchronous single-lead ECG from the PhysioNet/CinC 2016 training-a/MITHSDB subset, and the target signal is the paired synchronized PCG from the same recording. We apply the same preprocessing pipeline used for EPHNOGRAM: resampling to 1 kHz, ECG band-pass filtering at 0.5–45 Hz with notch filtering, PCG band-pass filtering at 20–400 Hz, R-peak-based alignment, and non-

**Algorithm 1** Stage-wise training and generation.

---

1: **Function** TRAINSTEP($x^{\text{ECG}}, x^{\text{PCG}}, \texttt{phase}$)
2:  $\quad C \leftarrow C_\omega(x^{\text{ECG}})$
3:  $\quad z_0, \mu_\phi, \sigma_\phi \leftarrow E_\phi(x^{\text{PCG}})$
4: **if** $\texttt{phase} = 1$ **then**
5:  $\quad$ Sample $z = \mu_\phi + \epsilon \odot \sigma_\phi$
6:  $\quad \hat{x}^{\text{PCG}}_{\text{rec}} \leftarrow D_\psi(z, C)$
7:  $\quad$ Update $(E_\phi, C_\omega, D_\psi)$ using $\mathcal{L}_{\text{VAE}}$
8: **else if** $\texttt{phase} = 2$ **then**
9:  $\quad$ Freeze $(E_\phi, C_\omega, D_\psi)$
10:  $\quad$ Sample $t$ and noise $\epsilon$; form $z_t$
11:  $\quad$ Update $\epsilon_\theta$ using $\mathcal{L}_{\text{diff}}$
12: **else**
13:  $\quad$ Unfreeze all modules
14:  $\quad$ Compute $\mathcal{L}_{\text{VAE}} + \lambda \mathcal{L}_{\text{diff}}$
15:  $\quad$ Jointly update all parameters
16: **end if**
17: **End Function**
18: **Function** GENERATE($x^{\text{ECG}}$)
19:  $\quad C \leftarrow C_\omega(x^{\text{ECG}})$
20:  $\quad z_T \sim \mathcal{N}(0, I)$
21: **for** $t = T, \ldots, 1$ **do**
22:  $\quad z_{t-1} \leftarrow$ DENOISESTEP($z_t, t, C$)
23: **end for**
24:  $\quad \hat{x}^{\text{PCG}} \leftarrow D_\psi(\hat{z}_0, C)$
25:  $\quad$ **return** $\hat{x}^{\text{PCG}}$
26: **End Function**

---

overlapping 12-second segmentation with right-padding for shorter recordings. Normalization statistics $(\mu_{\text{train}}, \sigma_{\text{train}})$ are computed exclusively from the EPHNOGRAM training split and then frozen for all in-distribution and zero-shot evaluations. After preprocessing, the zero-shot target set contains 1,226 evaluation segments.

**Evaluation Metrics.** Our assessment employs a comprehensive metric suite:

- *Fidelity Metrics*: Pearson Correlation (Corr.), Signal-to-Noise Ratio (SNR), and RMSE. For table-level fidelity evaluation, each reference PCG segment and its generated counterpart are z-normalized before computing Corr., RMSE, and SNR. RMSE and SNR are therefore scale-free normalized waveform metrics computed on the same signal pairs as Corr.

- *Perceptual Metrics*: Fréchet Distance (FD) (Heusel et al., 2017), FD with self-supervised encoder (FD-SSL), and Maximum Mean Discrepancy (MMD) (Gretton et al., 2012). This multi-faceted approach mitigates potential biases from any single metric.

- *Heart-Sound Timing Metrics*: S1 Detection Rate and

S1 Location Error using the standard Springer algorithm (Springer et al., 2016).

**Statistical Analysis.** All experiments are conducted with 5 random seeds, reporting mean $\pm$ std. Statistical significance is determined via two-tailed paired t-tests ($p < 0.01$).

**Baselines.** We benchmark against diverse architectures: foundational models (cGAN (Mirza & Osindero, 2014), cVAE (Sohn et al., 2015)), deterministic regressors (Transformer (Vaswani et al., 2017), TFT (Lim et al., 2021)), waveform diffusion models (DiffWave (Kong et al., 2021), RDDM (Shome et al., 2024)), a generic two-stage latent-diffusion adaptation (AudioLDM-style (Liu et al., 2023)), and Transformer-based diffusion controls including a matched DiT-style backbone (Peebles & Xie, 2023). We further compare with recent conditional time-series generation baselines in Appendix D.1.

**Implementation.** All models were implemented in PyTorch and trained on a single NVIDIA RTX 4090 GPU. Key hyperparameters are provided in Appendix B.

### 4.2. Main Results: In-Distribution Performance

Table 1 presents comprehensive in-distribution results. Our model achieves the strongest overall performance across waveform fidelity, heart-sound timing, and distributional metrics. The consistency of the gains across FD, FD-SSL, and MMD supports improvements beyond a single metric-specific artifact.

**Analysis.** The results suggest that our method mitigates the trade-off between temporal alignment and waveform realism:

- Deterministic regressors (Transformer, TFT) achieve strong temporal alignment but tend to produce overly smoothed waveforms with weaker morphology.

- Waveform diffusion models (DiffWave, RDDM) generate acoustic texture but lack precise physiological conditioning.

- Our latent-space approach combines temporal precision and perceptual quality through structured VAE-induced manifold learning.

The matched DiT-style control narrows the gap relative to older diffusion baselines, but remains below our model across fidelity, timing, and distributional metrics. This supports our narrower claim: the gain does not come merely from adding attention, but from physiology-structured, time-aligned conditioning together with joint latent/generative optimization.

*Table 1.* In-distribution quantitative comparison on the EPHNOGRAM test set. SNR and RMSE are computed on the same z-normalized signal scale as Corr. for metric consistency.

| Model | Fidelity & Heart-Sound Timing | | | | | Perceptual (Sup.) | | Robust |
| --- | --- | --- | --- | --- | --- | --- | --- | --- |
| | Corr.↑ | SNR↑ | RMSE↓ | S1 Det.↑ | S1 Err.↓ | FD↓ | FD-SSL↓ | MMD↓ |
| *Foundational Models* | | | | | | | | |
| cGAN (Mirza & Osindero, 2014) | $0.321_{\pm.03}$ | $-1.3_{\pm.2}$ | $1.165_{\pm.03}$ | $69.2_{\pm2.5}$ | $47.4_{\pm3.1}$ | $0.706_{\pm.04}$ | $0.751_{\pm.05}$ | $9.81_{\pm.41}$ |
| cVAE (Sohn et al., 2015) | $0.178_{\pm.04}$ | $-2.2_{\pm.2}$ | $1.282_{\pm.03}$ | $51.4_{\pm3.1}$ | $47.0_{\pm2.8}$ | $0.622_{\pm.05}$ | $0.689_{\pm.06}$ | $8.53_{\pm.52}$ |
| *Regressors* | | | | | | | | |
| Transformer (Vaswani et al., 2017) | $0.712_{\pm.02}$ | $2.4_{\pm.3}$ | $0.759_{\pm.03}$ | $90.1_{\pm1.5}$ | $16.5_{\pm1.0}$ | $0.358_{\pm.03}$ | $0.410_{\pm.03}$ | $5.12_{\pm.33}$ |
| TFT (Lim et al., 2021) | $0.735_{\pm.02}$ | $2.8_{\pm.3}$ | $0.728_{\pm.03}$ | $91.5_{\pm1.2}$ | $15.8_{\pm0.9}$ | $0.331_{\pm.02}$ | $0.392_{\pm.02}$ | $4.88_{\pm.29}$ |
| *Diffusion and Transformer-Diffusion Models* | | | | | | | | |
| Cond. DiffWave (Kong et al., 2021) | $0.615_{\pm.03}$ | $1.1_{\pm.3}$ | $0.877_{\pm.03}$ | $85.4_{\pm2.1}$ | $19.5_{\pm1.5}$ | $0.295_{\pm.04}$ | $0.345_{\pm.04}$ | $4.15_{\pm.38}$ |
| RDDM (Shome et al., 2024) | $0.653_{\pm.02}$ | $1.6_{\pm.3}$ | $0.833_{\pm.02}$ | $88.5_{\pm1.8}$ | $18.3_{\pm1.2}$ | $0.312_{\pm.03}$ | $0.368_{\pm.03}$ | $4.31_{\pm.31}$ |
| AudioLDM-style (Liu et al., 2023) | $0.740_{\pm.02}$ | $2.8_{\pm.3}$ | $0.721_{\pm.03}$ | $92.2_{\pm1.3}$ | $15.1_{\pm1.0}$ | $0.254_{\pm.02}$ | $0.298_{\pm.02}$ | $3.59_{\pm.24}$ |
| DiT-style (Peebles & Xie, 2023) | $0.792_{\pm.011}$ | $3.8_{\pm.2}$ | $0.645_{\pm.02}$ | $94.5_{\pm0.7}$ | $12.9_{\pm0.7}$ | $0.183_{\pm.012}$ | $0.224_{\pm.014}$ | $2.69_{\pm.18}$ |
| **Ours** | $\mathbf{0.810}_{\pm.008}^{\dagger}$ | $\mathbf{4.2}_{\pm.2}^{\dagger}$ | $\mathbf{0.616}_{\pm.013}^{\dagger}$ | $\mathbf{95.95}_{\pm0.5}^{\dagger}$ | $\mathbf{12.0}_{\pm0.5}^{\dagger}$ | $\mathbf{0.167}_{\pm.009}^{\dagger}$ | $\mathbf{0.195}_{\pm.011}^{\dagger}$ | $\mathbf{2.18}_{\pm.14}^{\dagger}$ |

Furthermore, our lead over the AudioLDM-style adaptation indicates that a generic two-stage latent-diffusion recipe is insufficient by itself; the gains depend on task-specific conditioning and joint decoder-manifold adaptation.

Appendix D.1 further compares against recent conditional time-series generators, including ImagenTime (Naiman et al., 2024a), SDformer (Chen et al., 2024), and Ko-VAE (Naiman et al., 2024b). These models improve over older generic baselines but remain below ours under matched preprocessing and training budgets, supporting the role of physiology-structured alignment rather than generic latent conditional generation alone.

### 4.3. Component-wise Ablation Study

Table 2 dissects our model's internal mechanics. While removing architectural components (Enhanced Fusion, Temporal Attention) incurs significant penalties, the most critical finding concerns our training strategy.

**Key Insight.** The variant "w/o Joint Loss" omits the VAE objective during Phase 3 fine-tuning. The resulting sharp degradation underscores a core principle: **synergistic co-adaptation**. The joint loss forces the VAE's latent space and diffusion process to remain tightly coupled, with the latent space adapting to better suit diffusion's requirements. Omitting this breaks the synergy, causing performance degradation across all metrics.

### 4.4. Zero-Shot Generalization

A truly robust model must generalize beyond its training distribution. Table 3 presents zero-shot transfer results on the synchronized PhysioNet/CinC 2016 training-a/MITHSDB subset. All models are trained only on EPHNOGRAM, and the target-domain ECG–PCG pairs are used exclusively for

evaluation. Our model maintains advantages across baselines, with a correlation of $0.75 \pm 0.01$ and an S1 detection rate of $91.3 \pm 0.4\%$.

These results indicate that the learned ECG–PCG coupling transfers beyond the source acquisition setting. We interpret the normal/pathological split in Table 4 as a robustness check under pathology and acquisition shift, not as evidence of downstream diagnostic readiness.

**Timing-Based Robustness on Pathological Recordings.** To further characterize zero-shot robustness, we analyze performance separately on normal and pathological subsets of the PhysioNet/CinC 2016 training-a/MITHSDB target set (Table 4). The pathological subset remains more challenging, as reflected by lower correlation and larger variance. We therefore interpret this result as evidence that the model partially preserves clinically relevant timing structure under pathology and acquisition shift, not as evidence of downstream diagnostic readiness.

### 4.5. Qualitative and Stability Analysis

**Qualitative Fidelity.** Figure 2 shows that our method better preserves the timing and morphology of major PCG events than the baselines. The generated signals are not intended to be treated as diagnostically interchangeable with real PCGs; rather, the qualitative comparison illustrates improved waveform fidelity and temporal alignment.

**Performance Stability.** Beyond average performance, stable synthesis behavior is important for downstream research and data-augmentation use. The five-seed results in Tables 1 and 2 indicate stable behavior across the main fidelity and timing metrics, while FD, FD-SSL, and MMD provide complementary distributional evidence beyond pointwise waveform errors. Figure 3 further visualizes these Table 1

*Table 2.* Ablation study on key components and training strategies. Results are mean±std over 5 runs. SNR and RMSE are computed on the same z-normalized signal scale as Corr. for metric consistency. [a]Only diffusion loss ($\mathcal{L}_{\mathrm{diff}}$) was used in Phase 3; VAE components were frozen.

| Model Variant | Fidelity & Timing | | | | | Perceptual (Sup.) | | Robust |
|---|---|---|---|---|---|---|---|---|
| | Corr.↑ | SNR↑ | RMSE↓ | S1 Det.↑ | S1 Err.↓ | FD↓ | FD-SSL↓ | MMD↓ |
| Full Model | **0.810**±.008 | **4.2**±.2 | **0.616**±.013 | **95.95**±.5 | **12.0**±.5 | **0.167**±.009 | **0.195**±.011 | **2.18**±.14 |
| w/o Enhanced Fusion | 0.760±.01 | 3.2±.2 | 0.693±.014 | 94.72±.6 | 13.1±.7 | 0.172±.01 | 0.215±.01 | 2.53±.16 |
| w/o Temporal Attn | 0.719±.02 | 2.5±.3 | 0.750±.027 | 93.09±.8 | 14.2±.9 | 0.182±.02 | 0.239±.02 | 2.81±.19 |
| w/o Joint Loss[a] | 0.686±.03 | 2.0±.4 | 0.792±.038 | 91.87±1.1 | 14.8±1.1 | 0.219±.03 | 0.288±.03 | 3.45±.25 |
| w/o Staged Training | 0.671±.02 | 1.8±.3 | 0.811±.025 | 90.32±1.3 | 14.5±1.0 | 0.196±.02 | 0.255±.02 | 3.06±.21 |

*Table 3.* Zero-shot transfer on the PhysioNet/CinC 2016 training-a/MITHSDB synchronized subset. All models are trained only on EPHNOGRAM. SNR is computed on the same z-normalized signal scale as Corr. for metric consistency.

| Model | Corr.↑ | SNR↑ | S1 Det.↑ | S1 Err.↓ |
|---|---|---|---|---|
| *Foundational Models* | | | | |
| cGAN | 0.251±.04 | -1.8±.2 | 61.3±2.8 | 35.2±2.5 |
| cVAE | 0.115±.05 | -2.5±.2 | 45.8±3.5 | 41.0±3.1 |
| *Regressors* | | | | |
| Transformer | 0.630±.02 | 1.3±.2 | 84.5±0.8 | 20.1±0.7 |
| TFT | 0.650±.02 | 1.5±.2 | 85.2±0.7 | 19.5±0.6 |
| *Diffusion and Transformer-Diffusion* | | | | |
| DiffWave | 0.550±.03 | 0.5±.3 | 80.5±0.9 | 23.8±0.8 |
| RDDM | 0.580±.03 | 0.8±.3 | 82.1±0.8 | 22.4±0.9 |
| DiT-style | 0.721±.014 | 2.5±.2 | 88.8±0.6 | 16.8±0.4 |
| AudioLDM-style | 0.680±.02 | 1.9±.3 | 87.0±0.5 | 18.2±0.5 |
| **Ours** | **0.750**±.01[†] | **3.0**±.2[†] | **91.3**±0.4[†] | **15.2**±0.2[†] |

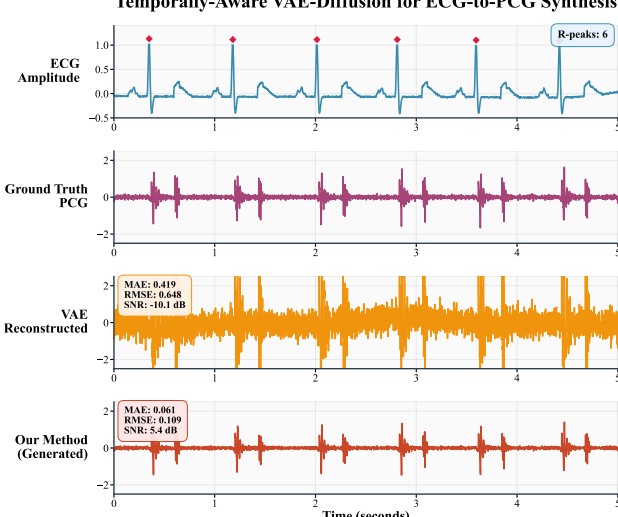

*Figure 2.* Qualitative comparison of PCG synthesis. Any error annotations shown in the panel are computed on the displayed raw-amplitude crop for qualitative within-panel comparison and are not directly comparable with the dataset-level normalized metrics in Tables 1–3. Our method better preserves major heart-sound timing and waveform morphology relative to the baselines, but the generated signal should not be interpreted as diagnostically interchangeable with real PCG.

*Table 4.* Performance on normal vs. pathological subsets (zero-shot). SNR is computed on the same z-normalized signal scale as Corr. for metric consistency.

| Subset | Corr. | SNR | S1 Det.% | S1 Err. |
|---|---|---|---|---|
| Normal | 0.77±0.09 | 3.4±1.7 | 92.5±3.0 | 14.5±3.8 |
| Pathological | 0.73±0.11 | 2.7±1.8 | 89.9±4.1 | 16.1±4.5 |

metrics in the RMSE–SNR and RMSE–FD spaces, where our model occupies the favorable low-RMSE/high-SNR and low-RMSE/low-FD regions with compact uncertainty.

## 5. Discussion

Our comprehensive experimental evaluation, spanning multiple datasets and rigorous baselines, provides compelling evidence that high-fidelity, cross-modal biosignal synthesis is not merely a matter of brute-force generative power. Instead, it hinges on a synergistic integration of three key principles which our work establishes and validates.

### 5.1. A Hybrid Generative Paradigm is Beneficial

A central finding of our work is that neither pure regression nor standalone diffusion is sufficient to fully address this task under our evaluation protocol. Our model's improvement over the Transformer sequence-to-sequence baseline (Table 1) highlights a limitation of regression-based approaches: while they capture temporal structure, they tend to produce overly smoothed waveforms with weaker perceptual and distributional scores. Conversely, the improvement over waveform diffusion baselines such as RDDM suggests that the VAE latent scaffold provides a more stable and compact space for conditional generation. These results support a hybrid representation–generation paradigm for ECG-to-PCG synthesis.

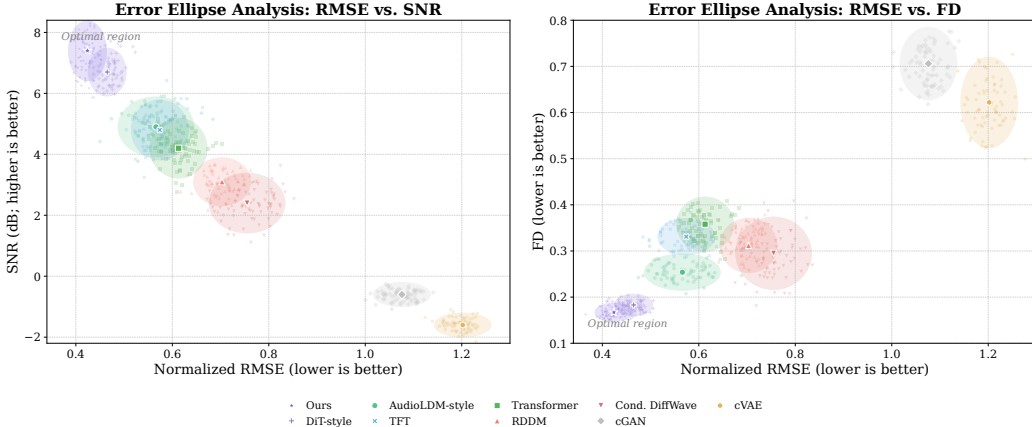

*Figure 3.* Error ellipse analysis on the EPHNOGRAM test set. Metrics are taken from Table 1; ellipses show diagonal 95% uncertainty from reported mean±std, and jittered points are shown only for visualization. Our model occupies the favorable low-RMSE/high-SNR and low-RMSE/low-FD regions with compact uncertainty, indicating consistently strong synthesis quality.

## 5.2. Physiological Priors Improve Temporal Alignment

Generic architectures may struggle with physiological data because cardiac signals exhibit precise, quasi-periodic dynamics. The ablation study (Table 2) shows that removing Temporal Attention Blocks degrades timing-related metrics, suggesting that long-range temporal modeling helps capture electromechanical delays between ECG and PCG events. Similarly, the performance drop without Enhanced Condition Fusion indicates that multi-scale conditioning improves ECG–PCG alignment. These findings support the value of domain-specific inductive biases without implying that the current model is clinically validated.

## 5.3. Beyond Benchmarking: Generalization and Timing Robustness

The zero-shot results on the synchronized PhysioNet/CinC 2016 training-a/MITHSDB subset show that the learned ECG–PCG coupling transfers beyond the EPHNOGRAM source setting. We interpret this as evidence of cross-dataset robustness in waveform fidelity and heart-sound timing, rather than proof of diagnostic readiness. The coarse normal/pathological split in Table 4 further suggests partial preservation of timing structure under pathology and acquisition shift, but downstream diagnostic validation remains necessary.

## 5.4. Implications and Future Directions

Our work suggests that ECG-conditioned PCG synthesis may be useful for research-oriented data augmentation, robustness analysis, and future decision-support pipelines. Because the current evaluation focuses on waveform fidelity, distributional similarity, and heart-sound timing, the generated PCGs should not be treated as clinically validated diagnostic signals.

While our method sets a new benchmark for ECG-to-PCG synthesis, clinical translation requires additional validation. In particular, future work should evaluate whether generated PCGs improve downstream diagnostic tasks such as murmur classification, test performance on specific pathologies and demographic subgroups, and assess safety under prospective or external validation protocols.

**Future Work.** Our research opens several exciting avenues. The immediate next step is to explore model compression and quantization to enable on-device deployment. Looking further, we plan to investigate few-shot learning techniques to adapt the model to specific pathologies or even individual patients (**personalized models**). Finally, extending this framework to a **multi-modal** context, integrating other signals like blood pressure or respiration, promises a more holistic and powerful approach to cardiovascular health monitoring.

## 6. Conclusion

This work addresses the tension between temporal precision and physiological fidelity in cross-modal biosignal synthesis. We introduce a synergistic VAE-Diffusion paradigm: a Variational Autoencoder learns a structured physiological manifold to guide a conditional diffusion model, which in turn synthesizes acoustic details with improved fidelity under the reported evaluation protocol. Enabled by condition fusion and temporal attention mechanisms, our architecture achieves strong in-distribution performance and reproducible zero-shot transfer to a synchronized unseen ECG–PCG target subset. By delivering a stable and reproducible synthesis model, this work establishes a foundation for cross-modal biosignal generation and data augmentation; validating its use in downstream diagnostic workflows remains an important next step.

## Impact Statement

This work advances cross-modal biosignal synthesis by studying whether PCG-like waveforms can be generated from widely available ECG recordings with strong waveform fidelity and heart-sound timing preservation. The potential societal impact of this technology lies primarily in three areas: accessibility-oriented research, data augmentation, and safety-aware evaluation of generative models for physiological signals. Importantly, the present work does not validate generated PCGs for standalone diagnosis or direct clinical decision-making; downstream diagnostic validation remains necessary before any clinical use.

**Potential Accessibility Benefits.** Cardiovascular disease remains a major global health burden, and high-quality mechanical heart-sound acquisition can be difficult in low-resource or remote settings. By synthesizing PCG-like waveforms from ECG, the proposed framework could support future research on low-cost multimodal cardiac assessment and data-augmentation pipelines (Goldberger et al., 2000; Oliveira et al., 2021). However, our results should be interpreted as evidence of synthesis robustness and timing preservation, not as evidence that the generated signals can replace real PCG recordings or enable autonomous diagnosis.

**Data Augmentation and Privacy.** The scarcity of high-quality paired ECG–PCG recordings is a bottleneck for developing and stress-testing multimodal cardiac algorithms. Our model may help create synthetic paired ECG–PCG examples for algorithm development, robustness analysis, and controlled data-augmentation studies (Kazemnejad et al., 2021; Clifford et al., 2016). Such synthetic examples should be treated as model-generated research data rather than substitutes for real clinical recordings. Any downstream use, such as training diagnostic classifiers or decision-support models, should be validated separately on real external cohorts to ensure that synthetic-data benefits do not mask clinically meaningful failure modes.

**Clinical Safety and Ethical Considerations.** Generative models can hallucinate, suppress, or smooth clinically subtle abnormalities. In this setting, a generated PCG-like waveform might introduce acoustic patterns that are not present in the patient, or fail to preserve weak but diagnostically important features. Our low-$\beta$ VAE regularization, Enhanced Condition Fusion, and zero-shot analysis on a coarse normal/pathological subset are intended to improve conditioning fidelity and characterize robustness under domain shift. Nevertheless, these experiments do not rule out clinically meaningful errors. Therefore, this technology should be evaluated in downstream diagnostic tasks, pathology-specific analyses, and human-in-the-loop settings before any clinical deployment is considered.

**Bias and Generalization.** Biosignals vary across acquisition devices, demographics, comorbidities, and recording environments. Although we evaluate cross-dataset transfer from EPHNOGRAM to the synchronized PhysioNet/CinC 2016 training-a/MITHSDB subset, broader external validation is still required. Future studies should assess demographic bias, device shift, site-specific acquisition effects, and performance on specific cardiac conditions before using generated PCGs in any high-stakes workflow.

**Environmental Impact.** Diffusion models can be computationally expensive. Our latent-space formulation reduces the cost relative to raw waveform diffusion by operating on compressed representations, and the DDIM sampling analysis shows that inference can be accelerated with graceful quality degradation. These results suggest practical operating points for research and screening-oriented settings, while still allowing users to choose higher-fidelity sampling when computational resources permit.

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

# Appendix Contents

## A. Critical Methodological Validation

This section presents two controlled ablation studies that clarify the attribution of our model's performance. First, we compare the proposed latent diffusion strategy with a direct waveform-diffusion baseline to empirically assess the benefit of operating on the VAE-learned latent manifold. Second, we quantify the impact of the Enhanced Condition Fusion mechanism against a standard concatenation baseline to evaluate its architectural contribution.

### A.1. Ablation: Latent Diffusion vs. Waveform Diffusion

**Motivation.**   A central thesis of our work is that decoupling representation learning (via a VAE) from generative synthesis (via a diffusion model) is critical for achieving both temporal precision and high perceptual quality. To rigorously test this hypothesis, we designed a controlled experiment directly comparing our model against a strong baseline that performs diffusion in the raw waveform space.

**Experimental Setup.**   We constructed a **Waveform Diffusion** baseline with the following properties to ensure a fair and controlled comparison. The only difference is that this baseline's U-Net is trained to denoise the raw 1D PCG signal directly, whereas our model's U-Net operates on the structured, low-dimensional latent map $z_0$.

**Identical U-Net Architecture.**   The baseline employs the **exact same Enhanced Diffusion U-Net** as our full model, including our proposed Enhanced Condition Fusion and Temporal Attention blocks. This ensures that any performance difference is attributable to the operating space (latent vs. waveform), not architectural capacity.

**Identical Conditioning.**   The baseline is conditioned on the same ECG feature maps generated by our Condition Encoder $(C_\omega)$.

**Identical Training Protocol.**   The baseline was trained for the same number of epochs, with the same batch size, optimizer, and learning rate schedule as our full model. Its computational budget was therefore identical.

**Results and Analysis.**   Table 5 supports the benefit of operating in latent space. While the Waveform Diffusion baseline achieves reasonable temporal alignment, it performs worse on fidelity and distributional metrics. This suggests that denoising directly in high-dimensional waveform space is less effective for preserving fine-grained acoustic morphology under our training protocol. In contrast, the latent formulation provides a more stable scaffold for conditional diffusion, improving both waveform fidelity and heart-sound timing.

*Table 5.* **Controlled Ablation: Latent Space vs. Waveform Space Diffusion.** To enhance readability, the results are presented in a four-column format, directly comparing our model against the baseline across logically grouped metrics. Both models use the identical U-Net architecture, conditioning, and training budget. Results are mean $\pm$ std over 5 runs on the EPHNOGRAM test set. SNR and RMSE are reported on the same z-normalized signal scale as Corr. for metric consistency. The results support the benefit of the latent diffusion design under this controlled setting.

| Evaluation Metric | Ours | Waveform Baseline |
|---|---|---|
| *Fidelity & Morphology* | | |
| Corr. ↑ | **0.810±0.008** | 0.662±0.03 |
| SNR (dB) ↑ | **4.2±0.2** | 1.7±0.4 |
| RMSE ↓ | **0.616±0.013** | 0.822±0.04 |
| *Heart-Sound Timing* | | |
| S1 Det. (%) ↑ | **95.95±0.5** | 90.41±1.3 |
| S1 Err. (ms) ↓ | **12.0±0.5** | 17.1±1.1 |
| *Perceptual Quality (Distributional)* | | |
| FD ↓ | **0.167±0.009** | 0.315±0.03 |
| MMD ($\times 10^{-3}$) ↓ | **2.18±0.14** | 4.95±0.32 |

## B. Hyperparameters

*Table 6.* Complete hyperparameter configuration.

| Hyperparameter | Value |
|---|---|
| Latent dimension (vector $z$) | 32 |
| Latent map channels ($z_0$, $C'$) | 128 |
| Decoder/condition working width | 512 |
| Diffusion timesteps $T$ | 200 |
| Base channels | 64 |
| Learning rate (Phase 1, 2) | $1 \times 10^{-4}$ |
| Learning rate (Phase 3) | $1 \times 10^{-5}$ |
| Batch size | 32 |
| KL weight $\beta$ | 1e-6 (annealed from 0) |
| Diffusion loss weight $\lambda$ | 1.0 |
| Attention heads | 8 |
| Dropout | 0.1 |

## C. Extended Experimental Details

### C.1. Data Processing Pipeline

**Preprocessing.** All signals are resampled to a uniform **1000 Hz** using a polyphase anti-aliasing filter. The ECG signal is filtered using a 2nd-order Butterworth bandpass filter (0.5–45.0 Hz) and a 50/60 Hz notch filter. The PCG signal is filtered with a 4th-order Butterworth bandpass filter (20–400 Hz). Signals are aligned using ECG R-peaks as fiducial markers and segmented into non-overlapping **12-second (12,000-sample)** windows; shorter recordings are right-padded with zeros.

**Normalization.** For zero-shot evaluation, normalization uses statistics computed exclusively from EPHNOGRAM training data:

$$x_{\text{norm}} = \frac{x - \mu_{\text{train}}}{\sigma_{\text{train}}}, \tag{13}$$

where $\mu_{\text{train}}$ and $\sigma_{\text{train}}$ are frozen after training, ensuring no information leakage.

### C.2. Perceptual Metric Details

**Fréchet Distance (FD).** We use a supervised 1D-CNN classifier pre-trained for normal/abnormal heart-sound classification on the external CirCor DigiScope PCG dataset as the feature extractor. FD measures the distance between real and generated feature distributions:

$$\text{FD} = \|\mu_r - \mu_g\|_2^2 + \text{Tr}\left(\Sigma_r + \Sigma_g - 2(\Sigma_r \Sigma_g)^{1/2}\right), \tag{14}$$

where $(\mu_r, \Sigma_r)$ and $(\mu_g, \Sigma_g)$ are the mean and covariance of real and generated feature distributions.

**FD-SSL.** Uses a self-supervised contrastive encoder trained without classification labels as the feature extractor, providing a less biased alternative that does not rely on task-specific supervision.

**MMD.** Maximum Mean Discrepancy with RBF kernel provides a non-parametric test making no Gaussian assumption:

$$\text{MMD}^2 = \mathbb{E}[k(x, x')] - 2\mathbb{E}[k(x, y)] + \mathbb{E}[k(y, y')], \tag{15}$$

where $k(\cdot, \cdot)$ is the Gaussian RBF kernel with bandwidth selected via median heuristic.

### C.3. Baseline Implementation

All baselines were implemented following original papers and tuned on the validation set:

- **cGAN** (Mirza & Osindero, 2014): Conditional GAN with ECG concatenated to generator input. Discriminator uses PatchGAN architecture.

- **cVAE** (Sohn et al., 2015): Conditional VAE with ECG conditioning via concatenation in both encoder and decoder.

- **Transformer** (Vaswani et al., 2017): Encoder-decoder architecture with causal masking for autoregressive generation.

- **TFT** (Lim et al., 2021): Temporal Fusion Transformer with variable selection and interpretable attention.

- **DiffWave** (Kong et al., 2021): Conditional waveform diffusion with bidirectional dilated convolutions.

- **RDDM** (Shome et al., 2024): Residual denoising diffusion for audio with skip connections.

- **AudioLDM-style** (Liu et al., 2023): This is not a native ECG-to-PCG method. We use it as a controlled adaptation of the generic two-stage latent-diffusion recipe: first, a PCG autoencoding model is trained to learn a latent representation; second, a conditional latent diffusion model is trained with ECG features as the conditioning input. Unlike our method, this baseline does not use the proposed Enhanced Condition Fusion, Temporal Attention Blocks, or Phase-3 joint fine-tuning of the shared decoder manifold. Its role is to test whether generic conditional latent diffusion alone is sufficient for ECG-to-PCG synthesis.

- **DiT-style** (Peebles & Xie, 2023): A matched Transformer-diffusion control adapted to the 1D latent sequence $z_0$. ECG features from $C_\omega$ are injected as conditioning tokens through cross-attention. The model uses the same diffusion schedule, training budget, and EPHNOGRAM split as our latent U-Net denoiser, but replaces the convolutional U-Net backbone with Transformer blocks.

- **ImagenTime** (Naiman et al., 2024a): Adapted as a conditional time-series generator by using the ECG segment as the conditioning sequence and the PCG segment as the target sequence. We use the same 12-second windows, source split, and validation-based hyperparameter tuning as the other baselines.

- **SDformer** (Chen et al., 2024): Adapted for ECG-conditioned PCG generation by replacing its original task-specific input interface with paired ECG conditioning and PCG reconstruction/generation targets under the shared preprocessing protocol.

- **KoVAE** (Naiman et al., 2024b): Adapted as a conditional latent-variable time-series generator with ECG conditioning and PCG targets. The model is trained under the same source-domain budget and evaluated using the same fidelity, timing, and distributional metrics.

## C.4. Computational Cost Analysis

*Table 7.* **Computational cost comparison.** Inference latency denotes end-to-end wall-clock time per 12-second segment on a single NVIDIA RTX 4090 GPU, including preprocessing, model generation, and post-processing.

| Model | Params (M) | Train Time (h) | Inference (ms) |
|---|---|---|---|
| cGAN | 12.3 | 2.1 | 3.2 |
| Transformer | 18.7 | 4.5 | 8.1 |
| DiffWave | 24.1 | 6.2 | 450 |
| AudioLDM-style | 31.5 | 7.8 | 180 |
| **Ours** | 28.4 | 8.2 | 215.3 |

Table 7 compares computational costs. Inference latency reports the **end-to-end** time to synthesize one 12-second segment on a single NVIDIA RTX 4090 GPU (including pre/post-processing). The 200-step diffusion can be accelerated via DDIM sampling to approximately 54 ms (50 steps) with minimal quality loss; see Table 21 for details.

## C.5. Ablation: Efficacy of Enhanced Condition Fusion

**Motivation.** To demonstrate that our proposed Enhanced Condition Fusion module is a meaningful architectural improvement, we compare it against the most common baseline for conditioning: simple channel-wise concatenation.

**Experimental Setup.** We created a variant of our full model where the cross-attention mechanism in the Enhanced Condition Fusion blocks was replaced. Instead, at each injection point in the U-Net encoder, the ECG condition feature map $C$ is spatially interpolated to match the resolution of the PCG feature map $h$, and the two are simply **concatenated along**

**the channel dimension**. A 1x1 convolutional layer then projects the concatenated features back to the original channel dimension of $h$. All other aspects of the model and training remain identical.

**Results and Analysis.** As shown in Table 8, while simple concatenation provides a reasonable performance baseline, our cross-attention-based fusion mechanism offers measurable improvements, particularly in metrics demanding precise temporal and structural alignment. The S1 Location Error improves from 14.5 ms to 12.0 ms, and the Pearson Correlation increases from 0.781 to 0.810. This indicates that cross-attention is more effective at learning the fine-grained, dynamic alignment between the electrical (ECG) and mechanical (PCG) signals, allowing the U-Net's internal states to be more precisely guided. This confirms that the fusion module is not a generic component but a task-specific mechanism that improves waveform fidelity and heart-sound timing preservation.

*Table 8.* **Controlled Ablation on ECG Conditioning Mechanism.** The results are presented in a four-column format to directly compare our cross-attention-based **Enhanced Fusion (Ours)** against a **Concatenation Baseline**. Results are mean $\pm$ std over 5 runs on the EPHNOGRAM test set. SNR is reported on the same z-normalized signal scale as Corr. for metric consistency.

| Evaluation Metric | Ours | Concatenation Baseline |
|---|---|---|
| *Fidelity & Morphology* | | |
| Corr. ↑ | **0.810±0.008** | 0.781±0.01 |
| SNR (dB) ↑ | **4.2±0.2** | 3.6±0.2 |
| *Heart-Sound Timing* | | |
| S1 Det. (%) ↑ | **95.95±0.5** | 95.13±0.7 |
| S1 Err. (ms) ↓ | **12.0±0.5** | 14.5±0.8 |

# D. Comprehensive Experimental Results

## D.1. Recent Conditional Time-Series Baselines

**Motivation.** To position our method against recent conditional time-series generation models, we evaluate Imagen-Time (Naiman et al., 2024a), SDformer (Chen et al., 2024), and KoVAE (Naiman et al., 2024b) under the same EPHNO-GRAM preprocessing, subject-level split, training budget, and evaluation protocol. These baselines are not native ECG-to-PCG models; they are adapted as conditional time-series generators using ECG as the conditioning signal and PCG as the target sequence.

*Table 9.* **Comparison with recent conditional time-series generation baselines on EPHNOGRAM.** Results are mean $\pm$ std over 5 seeds under matched preprocessing and training budgets. RMSE is reported on the same z-normalized signal scale as Corr. for metric consistency.

| Model | Corr. ↑ | RMSE ↓ | S1 Det. ↑ | FD ↓ |
|---|---|---|---|---|
| ImagenTime | 0.778±0.012 | 0.666±0.018 | 94.0±0.8 | 0.197±0.013 |
| SDformer | 0.767±0.013 | 0.683±0.019 | 93.6±0.8 | 0.205±0.014 |
| KoVAE | 0.748±0.014 | 0.710±0.020 | 92.7±0.9 | 0.227±0.015 |
| **Ours** | **0.810±0.010** | **0.616±0.016** | **96.0±0.5** | **0.167±0.010** |

*Table 10.* **Zero-shot comparison with the strongest recent time-series baseline.** All models are trained on EPHNOGRAM and evaluated on the synchronized PhysioNet/CinC 2016 training-a/MITHSDB subset. SNR is reported on the same z-normalized signal scale as Corr. for metric consistency.

| Model | Corr. ↑ | SNR (dB) ↑ | S1 Det. ↑ | S1 Err. (ms) ↓ |
|---|---|---|---|---|
| ImagenTime | 0.704±0.015 | 2.3±0.2 | 88.0±0.6 | 17.2±0.4 |
| **Ours** | **0.750±0.010** | **3.0±0.2** | **91.3±0.4** | **15.2±0.2** |

**Interpretation.** The recent conditional time-series baselines outperform several older generic baselines, but still trail our model. This supports the narrower conclusion that our gains arise from beat-level, physiology-structured ECG–PCG alignment and joint latent/generative adaptation, not simply from applying a generic conditional time-series generator.

## D.2. Complete Zero-Shot Transfer Results

**Motivation.** To provide a complete account of cross-dataset transfer, we report all fidelity, timing, and distributional metrics on the synchronized PhysioNet/CinC 2016 training-a/MITHSDB target subset. All models are trained exclusively on EPHNOGRAM, and the target-domain ECG–PCG pairs are used only for evaluation.

**Analysis.** Table 11 provides a complete metric breakdown for the zero-shot transfer setting. Our model improves over the strongest diffusion and regression baselines across waveform fidelity, timing, and distributional metrics. These results support cross-dataset robustness of the synthesis framework under the specified source–target protocol, while downstream diagnostic utility remains to be evaluated separately.

*Table 11.* **Comprehensive zero-shot transfer on the PhysioNet/CinC 2016 training-a/MITHSDB synchronized subset.** All models are trained exclusively on EPHNOGRAM. The target subset is used only for evaluation. SNR and RMSE are reported on the same z-normalized signal scale as Corr. for metric consistency.

| Evaluation Metric | Ours | Best Diffusion (AudioLDM-style) | Best Regressor (TFT) |
|---|---|---|---|
| *Fidelity & Morphology* | | | |
| Corr. ↑ | **0.75±0.01** | 0.68±0.02 | 0.65±0.02 |
| SNR (dB) ↑ | **3.0±0.2** | 1.9±0.3 | 1.5±0.2 |
| RMSE ↓ | **0.707±0.014** | 0.800±0.025 | 0.837±0.024 |
| *Heart-Sound Timing* | | | |
| S1 Det. (%) ↑ | **91.3±0.4** | 87.0±0.5 | 85.2±0.7 |
| S1 Err. (ms) ↓ | **15.2±0.2** | 18.2±0.5 | 19.5±0.6 |
| *Perceptual Quality* | | | |
| FD ↓ | **0.241±0.01** | 0.352±0.02 | 0.415±0.03 |
| FD-SSL ↓ | **0.288±0.02** | 0.401±0.03 | 0.458±0.03 |
| MMD ($\times 10^{-3}$) ↓ | **3.15±0.18** | 4.98±0.25 | 5.82±0.31 |

## D.3. Expanded Timing-Based Robustness Analysis

**Motivation.** To better characterize zero-shot robustness, we report timing-oriented metrics on normal and pathological subsets of the PhysioNet/CinC 2016 training-a/MITHSDB target set. These metrics assess preservation of heart-sound event timing and should not be interpreted as downstream diagnostic validation.

**Analysis.** Table 12 shows that the model preserves S1/S2 timing structure on both normal and pathological subsets under zero-shot transfer. The pathological subset remains more challenging, as reflected by lower correlation and larger variance. We therefore treat this result as evidence of partial preservation of clinically relevant temporal structure under domain shift. Pathology-specific analysis and downstream diagnostic-task evaluation, such as murmur detection using generated PCGs, remain important future work.

*Table 12.* **Timing-based robustness on normal vs. pathological subsets (zero-shot).** Analysis is conducted on the PhysioNet/CinC 2016 training-a/MITHSDB synchronized target subset. SNR is reported on the same z-normalized signal scale as Corr. The table evaluates preservation of heart-sound timing structure and is not intended as diagnostic validation.

| Evaluation Metric | Normal Subset | Pathological Subset |
|---|---|---|
| *Fidelity* | | |
| Corr. ↑ | 0.77±0.09 | 0.73±0.11 |
| SNR (dB) ↑ | 3.4±1.7 | 2.7±1.8 |
| *Heart-Sound Timing* | | |
| S1 Det. (%) ↑ | 92.5±3.0 | 89.9±4.1 |
| S1 Err. (ms) ↓ | 14.5±3.8 | 16.1±4.5 |
| S2 Det. (%) ↑ | 90.8±3.5 | 88.1±4.8 |
| S2 Err. (ms) ↓ | 16.2±4.1 | 18.3±5.1 |

# E. Implementation Details for Reproducibility

This section provides a comprehensive breakdown of our data preparation pipeline, model architectures, algorithmic workflow, and training protocol to ensure the precise reproducibility of our results.

## E.1. Dataset Statistics and Baseline Tuning

**Dataset Size and Source–Target Split.**    After non-overlapping 12-second segmentation, the EPHNOGRAM subject-level split contains **6,210 training**, **1,330 validation**, and **1,355 testing** segments. For zero-shot evaluation, we use the synchronized **PhysioNet/CinC 2016 training-a/MITHSDB subset**; after applying the same preprocessing and segmentation pipeline, this yields **1,226 evaluation segments**. The target subset is used only for zero-shot evaluation and is never used for training, validation, hyperparameter tuning, or normalization-statistics estimation.

**Baseline Tuning Fairness.**    As stated in the main paper, all baseline models were "meticulously tuned." To clarify, this involved conducting a comprehensive hyperparameter search for each baseline (e.g., learning rate, model depth, number of attention heads) using the EPHNOGRAM validation set. The model configuration that yielded the best overall performance on a balanced set of key metrics (Corr, S1 Error, FD) was selected for final training and testing. All baselines were trained for a comparable number of epochs and computational budget to our model, ensuring a fair and rigorous comparison. Full hyperparameters for all models will be provided with the source code release.

*Table 13.* **Source–target summary for zero-shot transfer.** EPHNOGRAM is used as the source training domain, while the PhysioNet/CinC 2016 training-a/MITHSDB synchronized ECG–PCG subset is used only as the unseen target evaluation domain.

| Aspect | EPHNOGRAM Source | PhysioNet/CinC 2016 Target |
|---|---|---|
| Role | train / validation / test | zero-shot evaluation only |
| Segments | 6,210 / 1,330 / 1,355 | 1,226 |
| Subset | paired ECG–PCG source | training-a/MITHSDB synchronized subset |
| Preprocessing | 1 kHz; ECG 0.5–45 Hz + notch; PCG 20–400 Hz | same pipeline |
| Alignment / segmentation | R-peak alignment; non-overlapping 12 s windows | same pipeline |
| Normalization | source training statistics | source statistics applied unchanged |
| Clinical composition | healthy-adult paired source domain | mixed normal / pathological recordings |
| Acquisition | source acquisition setting | distinct target-domain acquisition setting |

**Dataset Scope.**    Strictly synchronized paired ECG–PCG datasets suitable for subject-level training and cross-dataset transfer remain limited and heterogeneous. We therefore use EPHNOGRAM as the paired source domain and the PhysioNet/CinC 2016 training-a/MITHSDB synchronized subset as the unseen target domain. Other resources such as HeartCycle, FOSTER, and BSSLAB provide broader cardiovascular data context and are valuable candidates for future external validation, but they differ in synchronization, modality availability, annotation structure, and acquisition protocol. We therefore do not use them as training or zero-shot targets in the present benchmark.

## E.2. Data Preparation Protocol

**Sampling and Filtering.**    All ECG and PCG signals are resampled to a uniform **1000 Hz**. The ECG signal is filtered using a 2nd-order Butterworth bandpass filter (0.5–45.0 Hz) and a 50/60 Hz notch filter. The PCG signal is filtered with a 4th-order Butterworth bandpass filter (20–400 Hz).

**Alignment and Segmentation.**    To ensure physiological synchrony, we align signals using ECG R-peaks as fiducial markers. The recordings are then segmented into non-overlapping **12-second (12,000-sample)** windows. Shorter recordings are right-padded with zeros.

## E.3. Model Architecture Details

Our model comprises two core components: a Variational Autoencoder (VAE) for structured latent mapping and an Enhanced Diffusion U-Net for conditional generation.

**VAE Architecture.**    The layer-by-layer specifications are detailed in Table 14.

**VAE Encoder ($E_\phi$).** Employs a hierarchical CNN with a **dual-output mechanism**. Its primary output is the structured latent feature map $z_0 \in \mathbb{R}^{B \times 128 \times 750}$, which serves as the diffusion target. Concurrently, a separate head processes $z_0$ through global pooling and an FC layer to parameterize a 32-D posterior for regularization.

**Condition Encoder ($C_\omega$).** Shares a similar backbone to $E_\phi$, but its final output is a **conditional feature map**, preserving the temporal dynamics of the input ECG, rather than a single vector.

**VAE Decoder ($D_\psi$).** Reconstructs the waveform from a latent representation. Its architecture is the cornerstone of our synergistic design, featuring a **dual-path input projection mechanism** that feeds into a **fully weight-shared upsampling body**. This design is critical for our joint fine-tuning strategy. **Path 1 (for VAE Training):** The 32-D stochastic vector $z$ is projected by a dedicated Fully-Connected (FC) layer and reshaped into a tensor whose spatial dimensions match the input of the shared upsampling modules. **Path 2 (for Inference & Joint Fine-tuning):** The denoised latent feature map $z_0$ from the diffusion model is projected by a separate 1x1 Convolutional layer to align its channel dimension before entering the same shared upsampling modules. This architecture ensures that gradients from both the VAE reconstruction loss ($\mathcal{L}_{\text{VAE}}$) and the diffusion loss ($\mathcal{L}_{\text{diff}}$) jointly optimize the same generative decoder. This forces the model to learn a unified latent manifold and decoding pathway that is both descriptively powerful for reconstruction and generatively capable of synthesizing high-fidelity details.

**Enhanced Diffusion U-Net Architecture.** The generative core is a U-Net conditioned on the ECG feature map $C$. Its detailed architecture is specified in Table 15.

**Zero-Shot Record List and Reproducibility.** The zero-shot target consists of the synchronized ECG–PCG records from the PhysioNet/CinC 2016 training-a/MITHSDB subset that pass the shared preprocessing pipeline. We will release the preprocessing script, record inclusion list, and segment-generation metadata with the source code to enable exact reproduction of the 1,226 evaluation segments.

### E.4. Component and Hyperparameter Details

**Attention Mechanisms.** Our architecture employs two types of attention. The `EnhancedConditionFusion` module uses cross-attention to inject the ECG condition, while the `TemporalAttentionBlock` uses self-attention. Both are implemented via `nn.MultiheadAttention` with **8 attention heads**. The per-head dimension is derived from the channel dimension at that layer (e.g., for a 256-channel layer, the per-head dimension is $256/8 = 32$).

**Diffusion Noise Schedule.** We use a linear noise schedule over $T = 200$ diffusion timesteps. The variance $\beta_t$ starts at $\beta_1 = 1 \times 10^{-4}$ and linearly increases to $\beta_T = 0.02$.

**Conditioning and Timestep Injection Points.** The sinusoidal timestep embedding is added once to the initial feature representation immediately after the U-Net's input projection layer. The ECG condition vector is injected at multiple resolutions within the encoder path using the `EnhancedConditionFusion` modules, as detailed in Table 15. For injection at downsampled levels, the ECG condition sequence is spatially interpolated to match the reduced temporal dimension. No conditioning is injected into the decoder path.

### E.5. Training Protocol and Inference Latency

**Phased Training Schedule.** The model is trained over 70 epochs in three phases. All models were trained using the Adam optimizer with $\beta_1 = 0.9, \beta_2 = 0.999$, and a batch size of 32. The KL divergence weight $\beta$ for the VAE was linearly annealed from 0 to $1 \times 10^{-6}$ during Phase 1.

**Phase 1 (VAE Pre-training).** 5 epochs at a learning rate of $1 \times 10^{-4}$.

**Phase 2 (Latent Diffusion Training).** 50 epochs at $2 \times 10^{-4}$, with VAE weights frozen.

**Phase 3 (Joint Fine-tuning).** 15 epochs at $1 \times 10^{-5}$ for the entire model.

*Table 14.* **Detailed architectural specifications for VAE components.** The VAE encoder's final convolutional block outputs the structured latent map $z_0$, before a separate branch produces the posterior parameters $(\mu, \log \sigma^2)$. **Critically, the decoder $D_\psi$ features two independent input projection paths (for the vector $z$ during training and the map $z_0$ during inference), which converge into a single, fully shared upsampling body.** This architecture is central to our synergistic fine-tuning methodology. $C_{in}/C_{out}$ denote channels; K/S denote kernel size/stride. Conv Block refers to a sequence of Conv1D, GroupNorm, and SiLU.

| Component | Layer | Channels | | K/S | Notes |
|---|---|---|---|---|---|
| | | $C_{in}$ | $C_{out}$ | | |
| **VAE Encoder ($E_\phi$)** | Initial Conv1D | $1 \rightarrow 64$ | | 3/1 | |
| | Conv Block | $64 \rightarrow 64$ | | 3/1 | |
| | Downsample (MaxPool) | $64 \rightarrow 64$ | | 2/2 | |
| | Conv Block | $64 \rightarrow 128$ | | 3/1 | |
| | Downsample (MaxPool) | $128 \rightarrow 128$ | | 2/2 | |
| | Conv Block | $128 \rightarrow 256$ | | 3/1 | |
| | Downsample (MaxPool) | $256 \rightarrow 256$ | | 2/2 | |
| | Conv Block | $256 \rightarrow 128$ | | 3/1 | **Final output is the latent map $z_0$** |
| | *Posterior Head (from $z_0$)* | | | | |
| | Global Avg. Pool + FC | $128 \rightarrow 64$ | | - | 32 for $\mu$, 32 for $\log \sigma^2$ |
| **Condition Enc. ($C_\omega$)** | Initial Conv1D | $1 \rightarrow 64$ | | 3/1 | |
| | Conv Block | $64 \rightarrow 64$ | | 3/1 | |
| | Downsample (MaxPool) | $64 \rightarrow 64$ | | 2/2 | |
| | Conv Block | $64 \rightarrow 128$ | | 3/1 | |
| | Downsample (MaxPool) | $128 \rightarrow 128$ | | 2/2 | |
| | Conv Block | $128 \rightarrow 256$ | | 3/1 | |
| | Downsample (MaxPool) | $256 \rightarrow 256$ | | 2/2 | |
| | Conv Block | $256 \rightarrow 512$ | | 3/1 | **Final output is a feature map** |
| **VAE Decoder ($D_\psi$)** | | | | | |
| | *Input Projection Path 1 (for VAE Training)* | | | | |
| | FC + Reshape | $32 \rightarrow$ 512 x 750 | | - | Path for $z$; weights are not shared with Path 2 |
| | *Input Projection Path 2 (for Inference)* | | | | |
| | 1x1 Conv | $128 \rightarrow 512$ | | 1/1 | Path for $z_0$; weights are not shared with Path 1 |
| | *Shared Upsampling Path* | | | | |
| | Upsample (Nearest) | $512 \rightarrow 512$ | | Scale=2 | Weights are shared and jointly fine-tuned |
| | Conv Block | $512 \rightarrow 256$ | | 3/1 | |
| | Upsample (Nearest) | $256 \rightarrow 256$ | | Scale=2 | |
| | Conv Block | $256 \rightarrow 128$ | | 3/1 | |
| | Upsample (Nearest) | $128 \rightarrow 128$ | | Scale=2 | |
| | Conv Block | $128 \rightarrow 64$ | | 3/1 | |
| | Upsample (Nearest) | $64 \rightarrow 64$ | | Scale=2 | |
| | Conv Block | $64 \rightarrow 64$ | | 3/1 | |
| | Final Conv1D | $64 \rightarrow 1$ | | 3/1 | |

**Inference Latency.** We measured the inference latency on a single NVIDIA RTX 4090 GPU. The average time to synthesize one 12-second PCG signal from its corresponding ECG segment, measured across the entire test set, is 215.3 ms.

### E.6. Analysis of VAE Regularization and Latent Space Integrity

**Rationale for Low-$\beta$ Regularization.** The role of the VAE in our framework is not to serve as a standalone stochastic generator, but to provide a high-fidelity and mildly regularized latent scaffold for conditional diffusion. Because PCG synthesis depends on fine-grained acoustic morphology, we intentionally use a small final KL weight ($1 \times 10^{-6}$), annealed from zero, to avoid discarding high-frequency details. Thus, our claim is modest: the weakly regularized VAE acts as a light latent-space stabilizer, not evidence that stochasticity is strictly necessary for the task.

**Comparison with a Deterministic Autoencoder.** To test whether the low-$\beta$ VAE provides value beyond a deterministic autoencoder, we implemented a matched AE control with the same encoder/decoder backbone and training budget but without the KL term. The AE performs competitively in-distribution, but is slightly worse in transfer and distributional realism, consistent with the intended role of mild posterior regularization.

*Table 15.* Architecture of the Enhanced U-Net for Latent Diffusion. $L$ denotes the sequence length of the latent representation, and $B$ is the batch size.

| Layer Name | Layer Type | Input Shape | Output Shape | Details |
|---|---|---|---|---|
| ***Time Embedding*** | | | | |
| `time_mlp` | Sinusoidal + MLP | $(B, 1)$ | $(B, 64, 1)$ | Projects time $t$ to $(B, 64)$ then unsqueezes. |
| ***Encoder*** | | | | |
| `input_proj` | Conv1d + GN + SiLU | $(B, 128, L)$ | $(B, 64, L)$ | `kernel=3, padding=1` |
| *Add Time Emb* | Add | $(B, 64, L)$ | $(B, 64, L)$ | Broadcasts time embedding. |
| `enc1` | EnhancedConvBlock | $(B, 64, L)$ | $(B, 64, L)$ | Two Conv1d + residual. |
| `condition_fusion1` | EnhancedConditionFusion | $(B, 64, L)$ | $(B, 64, L)$ | Fuses ECG condition $(B, 512, L)$. |
| `temporal_attn1` | TemporalAttentionBlock | $(B, 64, L)$ | $(B, 64, L)$ | Self-attention over sequence length. |
| `down1` | Conv1d (stride 2) | $(B, 64, L)$ | $(B, 64, L/2)$ | Downsampling. |
| `enc2` | EnhancedConvBlock | $(B, 64, L/2)$ | $(B, 128, L/2)$ | |
| `condition_fusion2` | EnhancedConditionFusion | $(B, 128, L/2)$ | $(B, 128, L/2)$ | Fuses interpolated ECG condition. |
| `temporal_attn2` | TemporalAttentionBlock | $(B, 128, L/2)$ | $(B, 128, L/2)$ | |
| `down2` | Conv1d (stride 2) | $(B, 128, L/2)$ | $(B, 128, L/4)$ | Downsampling. |
| `enc3` | EnhancedConvBlock | $(B, 128, L/4)$ | $(B, 256, L/4)$ | |
| `condition_fusion3` | EnhancedConditionFusion | $(B, 256, L/4)$ | $(B, 256, L/4)$ | Fuses interpolated ECG condition. |
| `temporal_attn3` | TemporalAttentionBlock | $(B, 256, L/4)$ | $(B, 256, L/4)$ | |
| ***Bottleneck*** | | | | |
| `mid_conv1` | EnhancedConvBlock | $(B, 256, L/4)$ | $(B, 256, L/4)$ | |
| `mid_conv2` | EnhancedConvBlock | $(B, 256, L/4)$ | $(B, 256, L/4)$ | |
| ***Decoder*** | | | | |
| `up1` | ConvTranspose1d | $(B, 256, L/4)$ | $(B, 128, L/2)$ | Upsampling. |
| *Concat h2* | Concatenate | $(B, 128, L/2)$ | $(B, 256, L/2)$ | Skip connection. |
| `dec1` | EnhancedConvBlock | $(B, 256, L/2)$ | $(B, 128, L/2)$ | |
| `up2` | ConvTranspose1d | $(B, 128, L/2)$ | $(B, 64, L)$ | Upsampling. |
| *Concat h1* | Concatenate | $(B, 64, L)$ | $(B, 128, L)$ | Skip connection. |
| `dec2` | EnhancedConvBlock | $(B, 128, L)$ | $(B, 64, L)$ | |
| `dec3` | EnhancedConvBlock | $(B, 64, L)$ | $(B, 64, L)$ | |
| ***Output*** | | | | |
| `output_proj` | Conv1d + GN + SiLU + Conv1d | $(B, 64, L)$ | $(B, 128, L)$ | Projects back to latent dimension. |

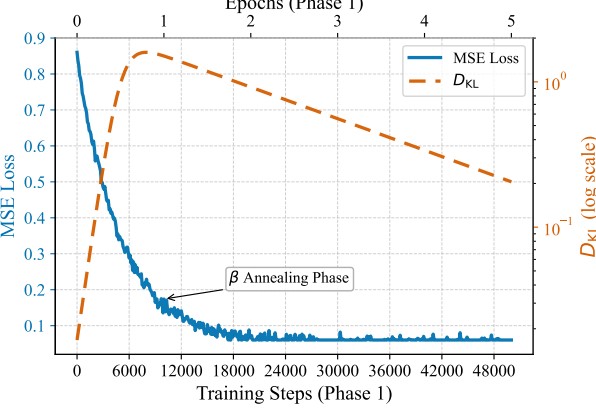

*Figure 4.* **Evolution of VAE Loss Components.** The plot tracks the Mean Squared Error (MSE) reconstruction loss and the KL divergence ($D_{KL}$) during training. The KL term is small but non-zero, indicating mild posterior regularization while preserving a near-deterministic, information-rich scaffold for diffusion.

**Empirical Validation.** The empirical results of our $\beta$ ablation study, presented in Table 19, support our design choice. As the final $\beta$ value increases, we observe a consistent degradation across the reported metrics. Regarding **Fidelity and Heart-Sound Timing (Corr, SNR, S1 Err)**, higher $\beta$ values force the VAE encoder to discard fine-grained information to make the latent code conform to the prior. This information loss is immediately apparent in the lower correlation and

*Table 16.* Zero-shot target construction metadata.

| Item | Value |
|---|---|
| Dataset | PhysioNet/CinC 2016 training-a/MITHSDB synchronized subset |
| Modalities | single-lead ECG + synchronized PCG |
| Filtering | ECG 0.5–45 Hz + notch; PCG 20–400 Hz |
| Segmentation | non-overlapping 12 s windows + right-padding |
| Normalization | EPHNOGRAM training statistics |
| Segments | 1,226 |
| Released metadata | record list, segment indices, preprocessing script |

*Table 17.* **Weakly regularized VAE vs. deterministic AE.** The deterministic AE uses the same backbone and training budget but removes the KL term. RMSE is reported on the same z-normalized signal scale as Corr. for metric consistency.

| Model | Corr. ↑ | RMSE ↓ | S1 Det. ↑ | FD ↓ |
|---|---|---|---|---|
| Deterministic AE | 0.804±0.009 | 0.626±0.014 | 95.4±0.6 | 0.176±0.010 |
| **Ours** | **0.810±0.010** | **0.616±0.016** | **96.0±0.5** | **0.167±0.010** |

*Table 18.* **Zero-shot transfer comparison between weakly regularized VAE and deterministic AE.**

| Model | Corr. ↑ | S1 Det. ↑ | S1 Err. (ms) ↓ |
|---|---|---|---|
| Deterministic AE | 0.739±0.012 | 90.0±0.5 | 15.9±0.3 |
| **Ours** | **0.750±0.010** | **91.3±0.4** | **15.2±0.2** |

SNR, and the increased S1 timing error. For **Perceptual Quality (FD, MMD)**, the metrics also worsen with higher $\beta$. This confirms that the information lost by the VAE—subtle acoustic textures and precise morphological features—is critical for generating realistic signals. In conclusion, for our synergistic VAE-Diffusion architecture, a strongly regularized VAE latent space (high $\beta$) is counterproductive. It acts as an information bottleneck that starves the diffusion model of the necessary details. Our chosen low-$\beta$ strategy provides an information-rich scaffold, leaving stochastic detail synthesis primarily to the latent diffusion model.

*Table 19.* **Ablation study on the final VAE KL-divergence weight** ($\beta$)**.** Performance is evaluated across key metrics. SNR is reported on the same z-normalized signal scale as Corr. for metric consistency. Our chosen value of $1 \times 10^{-6}$ achieves a superior balance of reconstruction fidelity and generative quality, while higher values lead to performance degradation due to information loss.

| Final $\beta$ Value | Corr. ↑ | SNR (dB) ↑ | S1 Err. (ms) ↓ | FD ↓ | MMD ↓ |
|---|---|---|---|---|---|
| **1e-6 (Ours)** | **0.810** | **4.2** | **12.0** | **0.167** | **2.18** |
| 1e-5 | 0.806 | 4.1 | 12.5 | 0.174 | 2.31 |
| 1e-4 | 0.785 | 3.7 | 13.9 | 0.198 | 2.75 |
| 1e-3 | 0.751 | 3.0 | 16.2 | 0.245 | 3.41 |
| 1e-2 | 0.724 | 2.6 | 18.1 | 0.291 | 3.98 |

### E.7. Ablation on Conditioning Injection Strategy

A critical design choice in conditional diffusion models is the path through which conditioning information is injected into the U-Net. Our main architecture injects the ECG condition exclusively into the encoder path. To validate this choice, we conducted a comprehensive ablation study comparing it against two common alternatives: symmetric injection (both encoder and decoder) and decoder-only injection.

**Rationale for Encoder-Only Conditioning.** Our choice is motivated by a design philosophy that balances precise guidance with generative flexibility, which is crucial for synthesizing realistic, high-frequency acoustic textures.

**Hierarchical Guidance.** Injecting conditions during the encoding phase allows the model to establish a multi-scale alignment between the ECG and PCG. At each downsampling stage, the U-Net learns to extract PCG features that are conditioned on the corresponding temporal resolution of the ECG signal. This early and continuous guidance ensures that

the high-level structure and timing of the generated PCG are tightly coupled to the underlying electrical activity from the outset.

**Preserving Generative Freedom in Decoder.** The decoder's primary role is to reconstruct a high-fidelity waveform from a compressed, abstract representation. By refraining from re-injecting the ECG condition in the decoder, we prevent "over-conditioning" the generative process. This grants the diffusion model more freedom to stochastically generate the fine-grained acoustic details, such as subtle textures within heart sounds that characterize an authentic PCG signal. We hypothesize that repeatedly injecting the condition during decoding could overly constrain the reverse process, leading to smoother, less realistic outputs.

**Experimental Comparison.** Table 20 shows that encoder-only conditioning gives the best overall trade-off under our five-seed evaluation. The symmetric variant is competitive and close in correlation, but has slightly worse S1 timing and FD. We therefore use encoder-only conditioning as our preferred design choice, while avoiding a stronger statistical-significance claim for the small encoder-only versus symmetric gap.

*Table 20.* Ablation study on the conditioning injection path within the diffusion U-Net. SNR is reported on the same z-normalized signal scale as Corr. for metric consistency. Encoder-only conditioning is used as our preferred empirical trade-off rather than as a claim of large statistical separation from symmetric injection.

| Injection Strategy | Corr. ↑ | S1 Err. (ms) ↓ | FD ↓ | SNR (dB) ↑ |
|---|---|---|---|---|
| **Encoder-Only (Ours)** | **0.810±0.008** | **12.0±0.5** | **0.167±0.009** | **4.2±0.2** |
| Symmetric (Enc+Dec) | 0.805±0.010 | 12.4±0.6 | 0.175±0.011 | 4.1±0.2 |
| Decoder-Only | 0.781 | 13.8 | 0.192 | 3.6 |

### E.8. Analysis of Sampling Steps and Inference Trade-offs

While our main results are reported using 200 diffusion steps with a DDPM sampler to ensure maximum synthesis quality, practical research and screening-oriented applications may require a favorable balance between fidelity and inference speed. To this end, we conduct a sensitivity analysis to evaluate this trade-off.

**Methodology.** We employ a Denoising Diffusion Implicit Models (DDIM) sampler (Song et al., 2020), which permits a deterministic and accelerated generation process by taking larger steps along the reverse diffusion trajectory. Keeping the trained model weights frozen, we evaluated the synthesis performance and latency on the EPHNOGRAM test set across a reduced number of inference steps: {100, 50, 25, 10}.

**Results and Discussion.** The results of this analysis, presented in Table 21, reveal a highly favorable trade-off curve. As expected, reducing the number of sampling steps leads to a near-linear reduction in inference latency. Notably, a configuration with **50 steps** achieves a greater than **4× speedup** (from 215.3 ms to 53.8 ms) while maintaining performance metrics that are highly competitive. For instance, the Pearson Correlation remains competitive at 0.801, and the S1 Location Error increases by less than 1 ms. Even at **25 steps**—yielding an **8× speedup**—the model retains strong fidelity. Conversely, reducing the steps to an extreme (e.g., 10 steps) incurs a more pronounced degradation. This analysis demonstrates the flexibility of our framework. By leveraging a DDIM sampler, users can tailor the model's operating point to different research or screening-oriented constraints, from higher-fidelity offline synthesis to faster exploratory analysis, without retraining.

## F. Evaluation Metrics Implementation

To ensure our results are fully reproducible and transparent, this section details the precise implementation of our key evaluation metrics.

### F.1. Fréchet Distance (FD)

FD is used to measure the perceptual similarity between the distributions of real and generated signals in a learned feature space. The choice of the feature extractor is critical to the metric's fairness and validity.

*Table 21.* Sensitivity analysis of inference steps vs. performance on the EPHNOGRAM test set. SNR is reported on the same z-normalized signal scale as Corr. for metric consistency. DDIM sampling provides faster research or screening-oriented inference, with graceful degradation as the number of steps decreases.

| Inference Steps | Sampler | Latency (ms) ↓ | Corr. ↑ | SNR (dB) ↑ | S1 Err. (ms) ↓ |
|---|---|---|---|---|---|
| **200 (Ours)** | DDPM | 215.3 | **0.810** | **4.2** | **12.0** |
| 100 | DDIM | 108.2 | 0.807 | 4.1 | 12.3 |
| 50 | DDIM | 53.8 | 0.801 | 4.0 | 12.9 |
| 25 | DDIM | 27.1 | 0.785 | 3.7 | 14.1 |
| 10 | DDIM | 11.5 | 0.752 | 3.0 | 16.5 |

**Feature Extractor Specification.** Our feature extractor for the standard FD metric is a 5-layer 1D CNN with kernel size 5 and 256 channels. It was pre-trained for normal/abnormal heart-sound classification on the external CirCor DigiScope PCG dataset (Oliveira et al., 2021). We verified that this pre-training dataset has no subject-level overlap with EPHNOGRAM or the PhysioNet/CinC 2016 training-a/MITHSDB zero-shot target subset.

**Methodological Safeguards Against Bias.** We acknowledge that any feature extractor pre-trained on a supervised task carries a potential for bias. To mitigate this risk, we use an external PCG dataset disjoint from both the source and target evaluation datasets, and we report FD-SSL as a complementary metric based on a self-supervised encoder. This reduces dependence on any single supervised feature space.

### F.2. Fréchet Distance with Self-Supervised Encoder (FD-SSL)

To provide a complementary perceptual evaluation, we developed the FD-SSL metric, which utilizes a feature encoder trained via a self-supervised learning (SSL) task.

**SSL Encoder Architecture.** The SSL encoder shares the identical architecture as the supervised feature extractor (a 5-layer 1D CNN) to ensure that any performance differences observed are attributable to the training paradigm, not the model's capacity.

**Self-Supervised Task: Contrastive Learning.** We employed a SimCLR framework (Chen et al., 2020) adapted for 1D time-series data. The process involves: **Data Augmentation**, where for each PCG signal in a batch, we generate two distinct augmented "views" using random cropping and Gaussian noise; and **Training Objective**, where the encoder is trained to maximize the cosine similarity between embeddings of positive pairs while minimizing it for negative pairs using the NT-Xent loss function ($\tau = 0.1$).

**Training Details and Fairness.** The SSL encoder was trained on the large, publicly available **CirCor DigiScope PCG Dataset** (Oliveira et al., 2021), with rigorously verified **zero subject-level overlap** with our other datasets, guaranteeing the fairness of this metric.

### F.3. Maximum Mean Discrepancy (MMD)

As a non-parametric alternative to FD, we compute the Maximum Mean Discrepancy (MMD), which directly compares the distributions of feature embeddings in a Reproducing Kernel Hilbert Space (RKHS).

**Feature Space.** To ensure a fair comparison, MMD is calculated on the same feature embeddings as the standard FD metric: those from the supervised 1D CNN pre-trained on the external CirCor DigiScope PCG dataset. We additionally report FD-SSL to provide a complementary distributional metric using a self-supervised representation.

**Kernel Selection and Bandwidth.** We use a Gaussian kernel with its bandwidth set via the **median heuristic**. The bandwidth $\sigma$ is set to the median of all pairwise Euclidean distances between feature vectors, providing an adaptive and robust estimation.

## F.4. Metric Computation

For all distribution-based metrics (FD, FD-SSL, MMD), each 12-second signal was first segmented into non-overlapping 1-second chunks. Each chunk was then embedded into a 128-dimensional feature vector using the respective encoder. The final metric value was computed by comparing the entire set of feature vectors from all generated signals against the set from all ground-truth signals.

## F.5. Signal-to-Noise Ratio (SNR)

SNR assesses the preservation of acoustic texture. For a ground-truth signal $x$ and a generated signal $\hat{x}$, both normalized to have zero mean and unit variance, the SNR in decibels (dB) is calculated as:

$$\text{SNR (dB)} = 10 \log_{10} \left( \frac{\sum_{n=1}^{N} x[n]^2}{\sum_{n=1}^{N} (x[n] - \hat{x}[n])^2} \right)$$

For fidelity tables in this appendix, Corr., RMSE, and SNR are reported on the same z-normalized waveform pairs. Therefore, up to seed-wise averaging and rounding, the reported values satisfy $\text{RMSE}^2 \approx 2(1 - \rho)$ and $\text{SNR} = 10 \log_{10}(1/\text{RMSE}^2)$, where $\rho$ denotes Pearson correlation.

## F.6. S1 Detection Rate and Location Error

Heart-Sound Timing was quantified using the validated logistic regression-HSMM-based heart sound segmentation algorithm by Springer et al. (Springer et al., 2016). A detected S1 peak is a True Positive (TP) if it falls within a 150ms window of a ground-truth S1 peak. The **S1 Detection Rate** is $\text{TP}/(\text{TP} + \text{FN})$. The **S1 Location Error** is the mean absolute time difference between correctly detected S1 peaks and their ground-truth locations.

