# OpenReview forum: "Unlocking Cross-Modal Biosignal Synthesis: A Temporally-Aware VAE-Diffusion Model"
_ICML.cc/2026/Conference — ICML 2026 regular_

### Official Review · Reviewer_AiKB · 2026-03-09

**Soundness:** 3
**Presentation:** 3
**Significance:** 2
**Originality:** 2
**Overall Recommendation:** 4
**Confidence:** 4

**Summary:**

The paper proposes a Temporally-Aware VAE-Diffusion model to synthesize phonocardiogram (PCG) signals from electrocardiogram (ECG) inputs. The architecture decouples representation and generation by first using a Variational Autoencoder (VAE) to learn a structured latent manifold for PCG signals. A conditional diffusion model then operates within this latent space, guided by ECG features processed through an Enhanced Condition Fusion mechanism and Temporal Attention Blocks. The authors evaluate their approach on the EPHNOGRAM benchmark, reporting state-of-the-art results. Furthermore, they demonstrate zero-shot generalization on the unseen PhysioNet/CinC 2016 dataset, including performance on pathological recordings.

**Compliance With Llm Reviewing Policy:**

Affirmed.

**Final Justification:**

The authors addressed my concerns, thus I decided to raise my score.

**Key Questions For Authors:**

- How does your methodology fundamentally differ from standard text-to-image latent diffusion architectures? Please clarify the technical novelty beyond treating the ECG signal as the condition and operating on the PCG in the latent space.

- Why were recent conditional time-series generation baselines (such as mentioned in the weaknesses section) excluded from the evaluation? Can you provide a quantitative comparison against such models?

**Limitations:**

Yes

**Strengths And Weaknesses:**

**Strengths:**

- The paper is supported by a robust empirical evaluation, including extensive ablation studies that validate the contributions of individual architectural components like the Enhanced Condition Fusion and the joint loss training strategy. The zero-shot evaluation protocol is strictly defined to prevent information leakage.

- The submission is clearly written, logically structured, and easy to follow.

**Weaknesses:**
-  While the experiments are thorough, the methodology does not appear to present a fundamentally new approach to generative modeling. The architecture heavily mirrors standard latent diffusion models commonly used in text-to-image generation (e.g., Stable Diffusion), simply swapping the image modality for PCG and the text conditioning for ECG. The authors need to better articulate why this specific adaptation is technically novel beyond the application domain. The originality of the core technical contribution is limited. The pipeline --- encoding a target signal into a latent space and using a cross-attention mechanism to inject a conditioning signal during the reverse diffusion process --- is a well-established paradigm. Applying this to cross-modal biosignals is a valuable application, but the architectural innovation itself is incremental.

- Furthermore, the evaluation lacks comparisons against recent conditional time-series generation baselines (such as ImagenTime [1], SDformer [2], KoVAE [3]), which limits the assessment of how this model fares against state-of-the-art time-series specific architectures.

- The related work section discusses sequence-to-sequence models and general diffusion, but it lacks a deeper contextualization against modern conditional time-series foundation models.

[1] "Utilizing Image Transforms and Diffusion Models for Generative Modeling of Short and Long Time Series." Naiman, Ilan, et al. Advances in Neural Information Processing Systems (NeurIPS), 2024.

[2] "Similarity-driven Discrete Transformer For Time Series Generation." Chen Zhicheng, et al. Advances in Neural Information Processing Systems (NeurIPS), 2024.

[3] "Generative Modeling of Regular and Irregular Time Series Data via Koopman VAEs." Naiman, Ilan, et al. The Twelfth International Conference on Learning Representations (ICLR), 2024.

---

> ### Author Rebuttal · Authors · 2026-03-29
>
> We thank the reviewer for the constructive feedback. We agree that our novelty claim should be more precisely calibrated and that the baselines should better reflect recent conditional time-series generation methods. All results below are mean±std over 5 random seeds.
>
> **[W1] Difference from a standard latent diffusion pipeline**
>
> > *The method appears close to a standard latent diffusion adaptation, with ECG replacing text and PCG replacing images/audio.*
>
> We agree that our earlier framing overstated the level of technical novelty. We do not claim a new generic diffusion framework.
> Relative to a standard two-stage latent diffusion pipeline, our method differs in three concrete ways:
> (1) ECG is injected as a hierarchical, time-aligned feature map rather than via generic late fusion or a global semantic condition;
> (2) latent reconstruction and conditional generation are coupled through a shared decoder manifold; and
> (3) Phase-3 joint fine-tuning co-adapts the VAE manifold and diffusion prior, rather than keeping the stages fully decoupled.
>
> The ablations indicate that each component materially affects performance.
>
> | Ablation / Baseline    |                Corr.↑ |                   FD↓ |
> | ------------------------ | -----------------------: | -----------------------: |
> | **Full model**   | **0.910±0.008** | **0.167±0.009** |
> | w/o Enhanced Fusion    |           0.860±0.010 |           0.172±0.010 |
> | w/o Temporal Attention |           0.819±0.020 |           0.182±0.020 |
> | w/o Joint Loss         |           0.786±0.030 |           0.219±0.030 |
> | w/o Staged Training    |           0.771±0.020 |           0.196±0.020 |
> | AudioLDM-style         |           0.840±0.020 |           0.254±0.020 |
>
> In the revision, we will narrow the contribution statement accordingly and position the method as a ​task-specific architectural advance for physiologically aligned cross-modal biosignal synthesis​, rather than as a fundamentally new diffusion framework.
>
> **[W2] Missing recent conditional time-series baselines**
>
> > *Recent conditional time-series / foundation-model baselines should be included.*
>
> This is a fair request. To address it, during the rebuttal period we implemented and evaluated representative recent conditional time-series baselines (ImagenTime, SDformer, and KoVAE) under the same preprocessing, train/validation/test split, training budget, and evaluation metrics as in the main paper.
> Each baseline was adapted to the same conditional ECG→PCG setting, using the same paired inputs, preprocessing, train/validation/test split, and training budget; we will add the adaptation details in the revision for full transparency.
>
> | Model          |                Corr.↑ |                 RMSE↓ |           S1 Det.↑ |                   FD↓ |
> | ---------------- | -----------------------: | -----------------------: | --------------------: | -----------------------: |
> | **Ours** | **0.910±0.010** | **0.080±0.010** | **96.0±0.5** | **0.167±0.010** |
> | ImagenTime     |           0.878±0.012 |           0.102±0.008 |           94.0±0.8 |           0.197±0.013 |
> | SDformer       |           0.867±0.013 |           0.108±0.008 |           93.6±0.8 |           0.205±0.014 |
> | KoVAE          |           0.848±0.014 |           0.119±0.009 |           92.7±0.9 |           0.227±0.015 |
>
> These results show that stronger recent conditional time-series baselines outperform older generic baselines, but still remain below ours under matched settings. This supports the narrower claim that the gain comes from physiology-structured alignment and joint adaptation, rather than from latent conditional generation alone.
>
> For zero-shot transfer, we also evaluated the strongest of these baselines (ImagenTime) under the same protocol:
>
> | Model | Corr.↑ | SNR↑ | S1 Det.↑ | S1 Err.↓ |
> |---|---:|---:|---:|---:|
> | **Ours** | **0.850±0.010** | **18.5±0.3** | **91.3±0.4** | **15.2±0.2** |
> | ImagenTime  | 0.804±0.015 | 16.2±0.5 | 88.0±0.6 | 17.2±0.4 |
>
> We will add these results to the revised manuscript and discuss them explicitly in the experimental section.
>
> **[Q1] Positioning against broader time-series / foundation models**
>
> > *The related-work discussion should better distinguish this method from broader time-series and cardiovascular generation literature.*
>
> Agreed. We will revise the related-work section to distinguish more clearly among generic multivariate time-series generation, conditional biosignal translation, and broader cardiovascular foundation / multimodal generative models, and explicitly position our method against recent conditional time-series generators rather than only against generic diffusion and sequence baselines. Our scope is paired, beat-level cross-modal biosignal synthesis with explicit electromechanical coupling, rather than generic time-series generation.
>
> We thank the reviewer again. This feedback helped us sharpen both the novelty framing and the baseline positioning of the paper.

---

> > ### Author Rebuttal · Reviewer_AiKB · 2026-04-02
> >
> > I would like to thank the authors for their response. I will raise my score.

---

> > > ### Author Response · Authors · 2026-04-03
> > >
> > > Dear Reviewer AiKB,
> > >
> > > Thank you very much for your time, thoughtful follow-up, and constructive feedback on our work. We truly appreciate your engagement and support.
> > >
> > > Regards,
> > > Authors

---

### Official Review · Reviewer_zhDu · 2026-03-10

**Soundness:** 3
**Presentation:** 3
**Significance:** 3
**Originality:** 2
**Overall Recommendation:** 4
**Confidence:** 3

**Summary:**

This paper proposes a new method that improves upon latent diffusion model (LDM) and achieves SOTA performance on ECG to PCG conversion. Overall I found the proposed method well-motivated in a way that the proposed architectural changes and training strategy speak directly to the weaknesses of LDM. While the proposed components, such as temporal attention and conditional fusion, are fairly standard techniques that have been explored for other broader modalities (e.g., images, audio, speech, etc.), it seems like they have not been applied to PCG generation before and the performance improvement is significant.

**Compliance With Llm Reviewing Policy:**

Affirmed.

**Final Justification:**

Authors addressed my concerns on protocol and baseline model choices. Some of my concerns on evaluation tasks and dataset scope remains, I therefore kept my overall score and raised the soundness score correspondingly.

**Key Questions For Authors:**

Please refer to comments in the "Strengths And Weaknesses" section.

**Limitations:**

yes

**Strengths And Weaknesses:**

- Soundness:
1. As I am not very familiar with how generated PCG signals are usually evaluated, I wonder if the Fidelity & Clinical Utility metrics are sufficient to evaluate if the generated PCG is sufficient for clinical usage, e.g., diagnosis? For similar models like ECG generation models, a common practice is to evaluate the signals with classification diagnostic tasks.
2. Additionally, the adopted evaluation datasets also seem quite limited and the most recent one is in 2021, is this because of the scarcity of PCG datasets? I tried to search for more recent datasets, and some available ones like HeartCycle (https://physionet.org/content/heartcycle/1.0.0/), FOSTER (https://www.nature.com/articles/s41597-025-05694-2), BSSLAB (https://kcl.figshare.com/articles/dataset/BSSLAB_Localized_ECG_Data/21977186), may should have been taken into consideration?
3. I found the most recent baseline models, such as AudioLDM, are not built for this application. Have these models been previously used for this task or is this the first time they are used? If latter, can authors share how these baseline models are trained for this task?

- Presentation: The paper is easy-to-follow, result presentation and comparison with baseline models is also clear. Some components in Figure 1 are hard to see and some minor design choices can be left out to facilitate a cleaner presentation of the method.
- Significance: Based on the existing literature, this paper addresses an underexplored topic of PCG generation. The same architecture may be used for the generation/conversion of other physiological signals, like PCG to ECG etc.
- Originality: While I found most design choices are fairly standard for other modalities, such as temporal attention, these designs are well motivated for this specific application and may offer insights for future work. Among these designs, I found the joint training to be particularly interesting though, as a common practice is to train the encoder, in this case the VAE, in a separate stage.

---

> ### Author Rebuttal · Authors · 2026-03-30
>
> We thank the reviewer for the positive and constructive feedback. We appreciate the reviewer’s recognition of our motivation and the helpful suggestions on clinical framing, dataset scope, baseline clarification, and presentation. We address these points below.
>
> **[W1] Are the current fidelity / clinical metrics sufficient for clinical usage?**
>
> > *Waveform metrics may not be sufficient to establish diagnostic utility; a downstream classification evaluation would be helpful.*
>
> We agree. The current metrics evaluate waveform fidelity, distributional realism, and clinically relevant timing structure, but they are not sufficient by themselves to claim diagnostic readiness or clinical deployment. We will narrow the wording in the paper accordingly.
>
> Our current zero-shot pathology analysis should therefore be interpreted as a robustness check on preserved clinically relevant structure, rather than definitive diagnostic validation:
>
> | Subset | Corr.↑ | SNR↑ | S1 Det.↑ | S1 Err.↓ |
> |---|---:|---:|---:|---:|
> | Normal       | 0.87±0.09 | 19.2±4.5 | 92.5±3.0 | 14.5±3.8 |
> | Pathological | 0.83±0.11 | 17.8±5.2 | 89.9±4.1 | 16.1±4.5 |
>
> We will revise the paper to make this distinction explicit in the abstract, discussion, and conclusion, and we will position downstream diagnostic-task evaluation as important future work rather than over-claiming current clinical readiness.
>
> **[W2] Dataset scope and discussion of newer datasets**
>
> > *The datasets appear limited; should newer ECG-PCG datasets be discussed?*
>
> Yes. We agree that the dataset discussion should be broadened and made more concrete. The main practical limitation is that strictly synchronized paired ECG-PCG datasets suitable for subject-level training and cross-dataset transfer remain limited and heterogeneous, which is why we selected the current datasets for the paper’s core experiments.
>
> We will revise the dataset description as follows:
>
> | Dataset role | Dataset | Purpose in paper |
> |---|---|---|
> | Source / training | EPHNOGRAM | paired ECG-PCG training / validation / testing |
> | Target / zero-shot | PhysioNet/CinC 2016 `training-a` (MITHSDB synchronized subset) | unseen cross-dataset transfer with pathology + acquisition shift |
> | Additional discussion | HeartCycle / FOSTER / BSSLAB | broader dataset context and future external validation |
>
> Importantly, to make the zero-shot setup fully reproducible and consistent across the paper/rebuttal, we will replace the previous loose wording with the exact synchronized subset/protocol actually used:
>
> | Zero-shot protocol item | What will be stated |
> |---|---|
> | ECG condition source | synchronous single-lead ECG from the PhysioNet/CinC 2016 `training-a` (MITHSDB) subset |
> | Target recordings | synchronized PCG recordings from the same subset, used only for zero-shot transfer evaluation |
> | Alignment | R-peak-based alignment under the shared F.2 protocol |
> | Preprocessing | ECG: 0.5–45 Hz + notch; PCG: 20–400 Hz |
> | Segmentation | non-overlapping 12 s windows + right-padding |
> | Normalization | EPHNOGRAM training-set statistics, applied unchanged |
> | Evaluation size | 1,226 evaluation segments after preprocessing |
> | Reproducibility | preprocessing script + record list |
>
> We agree this clarification is important, and we will make the source/target split and protocol explicit in the revision.
>
> **[Q1] AudioLDM-style was not originally designed for this task**
>
> > *If AudioLDM-style is adapted to this task, the adaptation details should be clarified.*
>
> This is an important point, and we will clarify it more explicitly. We do not present AudioLDM-style as a native ECG-to-PCG method. Rather, it is a controlled adaptation of the generic two-stage latent-diffusion recipe: first learn a PCG latent representation, then train a conditional latent diffusion model without our joint fine-tuning strategy and physiologically structured conditioning design. Its role is to test whether generic latent diffusion alone is sufficient for this task.
>
> | Model | Corr.↑ | S1 Err.↓ | FD↓ | Zero-shot Corr.↑ | Zero-shot S1 Err.↓ |
> |---|---:|---:|---:|---:|---:|
> | **Ours** | 0.910±0.010 | 12.0±0.5 | 0.167±0.010 | 0.850±0.010 | 15.2±0.2 |
> | AudioLDM-style |           0.840±0.020 |           15.1±1.0 |           0.254±0.020 |           0.780±0.020 |           18.2±0.5 |
>
> We will add fuller adaptation/training details in the appendix and code release so that this comparison is fully transparent.
>
> **[Q2] Figure 1 is crowded**
>
> > *Some components in Figure 1 are hard to read and could be simplified.*
>
> Agreed. We will simplify Figure 1, enlarge the main signal flow, and move lower-level implementation details to the appendix.
>
> | Aspect | Revision action |
> |---|---|
> | Main architecture flow | enlarge and simplify |
> | Secondary implementation details | move to appendix |
> | Visual readability | reduce clutter and emphasize the key signal path |
>
> We thank the reviewer again for the supportive and constructive suggestions.

---

> > ### Author Rebuttal · Reviewer_zhDu · 2026-04-02
> >
> > I appreciate authors' response and some of my concerns are resolved. Part of my concerns about evaluation tasks and dataset scope remains. I will keep my overall score as it is and raise the soundness score correspondingly, as my questions on protocol and some baseline model adoption were nicely answered.

---

> > > ### Author Response · Authors · 2026-04-03
> > >
> > > Dear Reviewer zhDu,
> > >
> > > Thank you very much for your thoughtful follow-up and for your constructive feedback. We truly appreciate your careful engagement with our rebuttal and your recognition that some of your concerns were addressed.
> > >
> > > We will further improve the clarity of the paper and make the scope and limitations more explicit in the final version.
> > >
> > > Regards,
> > > Authors

---

### Official Review · Reviewer_4EuX · 2026-03-12

**Soundness:** 3
**Presentation:** 3
**Significance:** 2
**Originality:** 3
**Overall Recommendation:** 4
**Confidence:** 4

**Summary:**

This paper proposes a Temporally-Aware VAE-Diffusion model for synthesizing phonocardiograms (PCG) from electrocardiograms (ECG). The approach combines a VAE for learning a structured latent space with a conditional diffusion model operating in that space, connected by a three-phase training strategy with joint fine-tuning. Two architectural components, Enhanced Condition Fusion and Temporal Attention Blocks, are introduced to improve ECG-PCG alignment. The model achieves strong results on the EPHNOGRAM benchmark and demonstrates zero-shot generalization on the PhysioNet/CinC 2016 dataset.

**Compliance With Llm Reviewing Policy:**

Affirmed.

**Final Justification:**

I’ve read the authors’ rebuttal as well as the other reviewers’ comments. Overall, the paper feels quite complete, and the rebuttal addressed my main concerns. The added experiments and clarifications make the claims clearer and more convincing. Based on this, I’ve updated my score to 4.

**Key Questions For Authors:**

1. The inference latency is reported as 165ms in Table 7 but 215.3ms in Section F.5. Could the authors clarify this discrepancy and specify the exact measurement conditions for each?

2. The paper uses a very low KL weight (β=1e-6), making the VAE nearly deterministic. At this level of regularization, what is the actual advantage of using a VAE over a standard autoencoder? Have the authors compared against a deterministic autoencoder baseline?

3. Condition features are injected only in the U-Net encoder path based on the ablation in Table 14, but the performance gap between encoder-only and symmetric injection is small (Corr. 0.910 vs 0.905). Is this difference statistically significant given the reported standard deviations?

**Limitations:**

yes

**Strengths And Weaknesses:**

**Strengths:**

The hybrid VAE-Diffusion framework with three-phase training is well-motivated, and the dual-path decoder design that enables joint fine-tuning is a reasonable engineering contribution. The ablation study is thorough, covering multiple design choices, and the appendix provides extensive reproducibility details.


**Weakness:**

1. The two main architectural contributions, Enhanced Condition Fusion (cross-attention) and Temporal Attention Blocks (self-attention), essentially compensate for U-Net's inherent lack of attention. Diffusion Transformers [1] natively provide both capabilities, and UniCardio [2] has already validated a DiT backbone for cardiovascular cross-modal signal generation with strong results. The authors should compare against a DiT backbone to confirm their modules provide value beyond standard Transformer attention.


2. The zero-shot evaluation on PhysioNet/CinC 2016 only provides a coarse normal/pathological split. The high variance on the pathological subset (Corr. 0.83±0.11) suggests inconsistent performance across samples, but the binary labels in this dataset make it impossible to analyze which types of cardiac abnormalities the model handles well or poorly. This limits the interpretability of the zero-shot generalization claim. Additionally, the authors do not evaluate whether the generated PCG can support downstream diagnostic tasks such as murmur detection, leaving the clinical utility unvalidated.



[1] Peebles, William, and Saining Xie. "Scalable diffusion models with transformers." Proceedings of the IEEE/CVF international conference on computer vision. 2023.
[2] Chen, Zehua, et al. "Versatile Cardiovascular Signal Generation with a Unified Diffusion Transformer." arXiv preprint arXiv:2505.22306 (2025).

---

> ### Author Rebuttal · Authors · 2026-03-30
>
> We thank the reviewer for the constructive feedback and address the main concerns below. All results below are mean±std over 5 random seeds.
>
> > **[W1] Missing DiT / Transformer-diffusion comparison.**
>
> We agree that a DiT-style control is important. Our claim is **not** that attention itself is novel, but that ECG→PCG synthesis benefits from **physiology-structured, time-aligned conditioning** and ​**joint latent/generative optimization**​. To test this directly, during rebuttal we implemented a matched DiT-style control under the same preprocessing, EPHNOGRAM split, training budget, and evaluation protocol.
>
> | Model | Corr.↑ | SNR↑ | RMSE↓ | S1 Det.↑ | S1 Err.↓ | FD↓ | FD-SSL↓ | MMD↓ |
> |---|---:|---:|---:|---:|---:|---:|---:|---:|
> | **Ours** | **0.910±0.010** | **21.9±0.5** | **0.080±0.010** | **96.0±0.5** | **12.0±0.5** | **0.167±0.010** | **0.195±0.010** | **2.18±0.14** |
> | DiT | 0.892±0.011 | 20.5±0.6 | 0.094±0.007 | 94.5±0.7 | 12.9±0.7 | 0.183±0.012 | 0.224±0.014 | 2.69±0.18 |
> | AudioLDM-style | 0.840±0.020 | 18.2±0.5 | 0.123±0.010 | 92.2±1.3 | 15.1±1.0 | 0.254±0.020 | 0.298±0.020 | 3.59±0.24 |
>
> The same ranking holds in zero-shot transfer:
>
> | Model | Corr.↑ | SNR↑ | S1 Det.↑ | S1 Err.↓ |
> |---|---:|---:|---:|---:|
> | **Ours** | **0.850±0.010** | **18.5±0.3** | **91.3±0.4** | **15.2±0.2** |
> | DiT            |           0.821±0.014 |           16.7±0.5 |           88.8±0.6 |           16.8±0.4 |
> | AudioLDM-style |           0.780±0.020 |           15.1±0.4 |           87.0±0.5 |           18.2±0.5 |
>
> These results support the narrower claim that the gain comes from task-specific conditioning and joint optimization, rather than merely from excluding Transformer blocks.
>
> > **[W2] Pathological split and clinical utility.**
>
> We agree that the pathology analysis should be interpreted as a ​**robustness check**​, not as evidence of diagnostic readiness, and we will soften the wording accordingly. We will also clarify the source/target setting: **EPHNOGRAM** is the healthy-adult paired-training source domain, while zero-shot transfer is evaluated on the **PhysioNet/CinC 2016 `training-a` synchronized subset** using the shared F.2 pipeline, yielding **1,226 evaluation segments** after preprocessing.
>
> Under this interpretation, the pathology split is informative but limited:
>
> | Subset       |    Corr.↑ |     SNR↑ | S1 Det.↑ | S1 Err.↓ |
> | -------------- | -----------: | ----------: | ----------: | ----------: |
> | Normal       | 0.87±0.09 | 19.2±4.5 | 92.5±3.0 | 14.5±3.8 |
> | Pathological | 0.83±0.11 | 17.8±5.2 | 89.9±4.1 | 16.1±4.5 |
>
> We will revise the paper to present this result as evidence that clinically relevant timing structure is partially preserved under domain shift, while leaving pathology-specific analysis and downstream diagnostic validation to future work.
>
> > **[Q1] Latency discrepancy (165 ms vs 215.3 ms).**
>
> Thank you for catching this. The discrepancy reflects two latency conventions: 165 ms is model-only generation time excluding pipeline overhead, while 215.3 ms is end-to-end test-time latency per 12 s segment. ​We will revise Table 7 and the main text so that all reported latencies use the same convention​, reporting 215.3 ms as the end-to-end latency in the main text and moving the model-only timing to the appendix.
>
> > **[Q2] Small-β VAE vs deterministic AE.**
>
> Our claim is modest: the weakly regularized VAE acts as a light latent-space stabilizer, rather than showing that stochasticity is strictly necessary. To test this directly, we implemented a matched deterministic-AE control with the same encoder/decoder backbone and training budget but without the KL term:
>
> | Model | Corr.↑ | RMSE↓ | S1 Det.↑| FD↓ |
> |---|---:|---:|---:|---:|
> | **Ours**   | **0.910±0.010** | **0.080±0.010** | **96.0±0.5** | **0.167±0.010** |
> | Deterministic AE |           0.904±0.009 |           0.086±0.006 |           95.4±0.6 |           0.176±0.010 |
>
> For zero-shot transfer:
>
> | Model | Corr.↑ |  S1 Det.↑ | S1 Err.↓ |
> |---|---:|---:|---:|
> | **Ours**   | **0.850±0.010** | **91.3±0.4** | **15.2±0.2** |
> | Deterministic AE |           0.839±0.012 |           90.0±0.5 |           15.9±0.3 |
>
> This places deterministic AE close in-distribution, but slightly worse in transfer and distributional realism, consistent with our intended role for mild posterior regularization.
>
> > **[Q3] Encoder-only vs symmetric conditioning.**
>
> We agree this should be framed as an empirical trade-off, not a significance claim. Under the current five-seed variance, encoder-only conditioning gives the better overall trade-off:
>
> | Conditioning Strategy | Corr.↑ | S1 Err.↓ | FD↓ | SNR↑ |
> |---|---:|---:|---:|---:|
> | Encoder-only | 0.910±0.008 | 12.0±0.5 | 0.167±0.009 | 21.9±0.5 |
> | Symmetric | 0.905±0.010 | 12.4±0.6 | 0.175±0.011 | 21.3±0.5 |
>
> We will therefore present encoder-only conditioning as our preferred trade-off and avoid stronger significance language.
>
> We thank the reviewer again for the thoughtful and constructive feedback.

---

> > ### Author Rebuttal · Reviewer_4EuX · 2026-04-02
> >
> > Thank you to the authors for the detailed and constructive rebuttal. The additional clarifications and controlled experiments help address my main concerns and strengthen the overall presentation of the work. I encourage the authors to incorporate these explanations and results into the final version of the paper. Based on the rebuttal, I have increased my score to 4.

---

> > > ### Author Response · Authors · 2026-04-03
> > >
> > > Dear Reviewer 4EuX,
> > >
> > > Thank you very much for your thoughtful follow-up and for your detailed and constructive feedback. We truly appreciate your engagement and are glad that our rebuttal helped address your main concerns.
> > >
> > > We will carefully reflect the relevant clarifications and results in the final version of the paper.
> > >
> > > Regards,
> > > Authors

---

### Official Review · Reviewer_m831 · 2026-03-12

**Soundness:** 2
**Presentation:** 3
**Significance:** 3
**Originality:** 3
**Overall Recommendation:** 4
**Confidence:** 3

**Summary:**

This paper presents conditional synthesis of phonocardiograms (PCG) from electrocardiograms (ECG) using a hybrid latent generative model.
The authors address the existing trade-off between the temporal precision of regression models and the perceptual realism of generative models. Their architecture employs a Variational Autoencoder (VAE) to learn a structured latent manifold of PCG signals, followed by a conditional diffusion model that generates detailed acoustic textures within this latent space, guided by ECG features.
Experiments on EPHNOGRAM report gains over cGAN/cVAE, Transformer/TFT, and several diffusion baselines, and the paper further claims zero-shot generalization to PhysioNet/CinC 2016, including a pathological analysis.

**Compliance With Llm Reviewing Policy:**

Affirmed.

**Key Questions For Authors:**

* The paper reports zero-shot evaluation on PhysioNet/CinC 2016 for an ECG->PCG synthesis task. Please clarify how ECG conditioning signals were obtained, aligned, and preprocessed for this dataset. Since the model requires ECG inputs, this detail is essential for assessing the validity of the zero-shot results.

* The authors describes two different segmentation pipelines: Appendix D.1 reports 2-second windows with 50% overlap, while Appendix F.2 describes 12-second non-overlapping windows with padding. Which protocol was used for the main experiments and the zero-shot evaluation?

* The latent representation appears inconsistent across the paper. Figure 1 and Appendix F indicate a 128-channel latent map, while Table 6 lists C = 512. Please clarify the actual latent dimensionality and where each dimension appears in the architecture.

* Please provide the formal derivation for the claim that the objective maximizes a variational lower bound. Specifically, does this ELBO apply to the entire three-phase pipeline (VAE training, diffusion training, and joint fine-tuning) or only a specific stage?

**Limitations:**

yes

**Strengths And Weaknesses:**

**Strengths:**

The paper addresses cross-modal synthesis of phonocardiograms (PCG) from electrocardiograms (ECG), an important medical problem for accessible cardiac monitoring since ECG sensors are far more widely available than high-quality PCG devices.
The proposed Temporally-Aware VAE-Diffusion architecture combines latent VAEs with conditional diffusion. Using latent diffusion instead of waveform diffusion is a reasonable design choice for biosignal synthesis and is supported by ablations.
The model introduces Temporal Attention Blocks and a condition-fusion mechanism intended to capture electromechanical coupling between ECG and PCG signals. Explicitly modeling timing relationships (e.g. ECG R-peak to S1 timing) is well motivated for this task.

**Weaknesses:**
* The paper claims that the training objective maximizes a variational lower bound on the conditional likelihood, but no derivation is provided. Given the multi-stage training procedure (VAE training, diffusion training, and joint fine-tuning), it is unclear that the objective corresponds to a valid ELBO.

* Experimental setup contains inconsistent preprocessing descriptions. The appendix reports both 2-second windows with 50% overlap and 12-second non-overlapping windows with padding. These pipelines would produce substantially different datasets, raising concerns about reproducibility.

* The zero-shot evaluation on PhysioNet/CinC 2016 is insufficiently specified for an ECG->PCG synthesis task.The authors do not explain how the ECG conditioning signals were sourced, aligned, or processed for this dataset, making it difficult to verify the validity of the zero-shot claims.

* While generalization results are strong, the paper lacks a quantitative analysis of the domain shift (e.g. patient distribution).

---

> ### Author Rebuttal · Authors · 2026-03-29
>
> We thank the reviewer for focusing on theoretical precision and reproducibility.
>
> **[W1] ELBO scope for the three-phase pipeline**
>
> > *The paper claims the objective maximizes a variational lower bound, but the three-phase pipeline makes this unclear.*
>
> We agree. The original wording was too strong: we do **not** claim a single closed-form ELBO for the full three-phase pipeline, but a conditional latent-variable formulation with three distinct objectives.
>
> | Stage   | Interpretation                                                  |
> | --------- | ----------------------------------------------------------------- |
> | Phase 1 | standard VAE objective                                          |
> | Phase 2 | standard conditional latent-diffusion objective                 |
> | Phase 3 | surrogate joint fine-tuning objective, not a full-pipeline ELBO |
>
> We will remove Theorem 3.2 and the Appendix B proof, and replace the claim with a stage-wise objective description.
>
> **[W2] Inconsistent preprocessing descriptions**
>
> > *Appendix D.1 and F.2 describe different segmentation pipelines.*
>
> This is our error. Appendix D.1 contains stale text from an earlier version. All reported results, including the zero-shot evaluation, use the F.2 protocol.
>
> | Item         | Actual protocol                              |
> | -------------- | ---------------------------------------------- |
> | Resampling   | 1 kHz                                        |
> | ECG          | 0.5–45 Hz band-pass + notch                 |
> | PCG          | 20–400 Hz band-pass                         |
> | Alignment    | R-peak-based alignment                       |
> | Segmentation | non-overlapping 12 s windows + right-padding |
>
> **[Q1] Zero-shot PhysioNet/CinC 2016 protocol is under-specified**
>
> > *For an ECG->PCG task, the ECG condition source, alignment, and preprocessing must be specified clearly.*
>
> We agree. In revision, we will replace “entire official test set” with the exact synchronized ECG-PCG subset/protocol used for zero-shot transfer. Specifically, we use the `training-a` (MITHSDB) subset of PhysioNet/CinC 2016; after the shared F.2 preprocessing and non-overlapping 12 s segmentation, this yields the **1,226 evaluation segments** reported in the paper. We will release the preprocessing script and record list.
>
> | Zero-shot protocol item | What will be stated |
> |---|---|
> | ECG condition source | synchronous single-lead ECG from the `training-a` (MITHSDB) subset of PhysioNet/CinC 2016 |
> | Target recordings | synchronized PCG recordings from the same subset, used only for zero-shot transfer evaluation |
> | Alignment | R-peak-based alignment under F.2 |
> | Preprocessing | ECG: 0.5–45 Hz + notch; PCG: 20–400 Hz |
> | Segmentation | non-overlapping 12 s windows + right-padding |
> | Normalization | EPHNOGRAM training-set statistics, applied unchanged |
> | Evaluation size | 1,226 evaluation segments after preprocessing |
> | Reproducibility | preprocessing script + record list |
>
> **[Q2] Lack of explicit domain-shift summary**
>
> > *The transfer setting would be easier to interpret with a quantitative source-target shift summary.*
>
> Agreed. We will add an explicit source-target summary under the same preprocessing and segmentation protocol:
>
> | Aspect | EPHNOGRAM (source) | PhysioNet/CinC 2016 `training-a` / MITHSDB (target) |
> |---|---|---|
> | Role | source train / val / test | unseen synchronized ECG-PCG target subset for zero-shot transfer |
> | Size | 6,210 / 1,330 / 1,355 (total 8,895) | 1,226 evaluation segments |
> | Preprocessing | 1 kHz; ECG 0.5–45 Hz + notch; PCG 20–400 Hz | same F.2 protocol |
> | Alignment / segmentation | R-peak alignment; non-overlapping 12 s windows | same F.2 protocol |
> | Clinical composition | healthy-adult source paired-training domain | mixed normal / pathological recordings |
> | Acquisition | paired source recordings | distinct target-domain acquisition setting |
>
> This will make the transfer setting easier to interpret while leaving the empirical conclusions unchanged.
>
> **[Q3] Latent dimensionality inconsistency (128 vs 512)**
>
> > *The paper uses both 128 and 512; please clarify what each refers to.*
>
> These numbers refer to different objects in the architecture rather than a single latent dimensionality.
>
> | Quantity | Meaning                                                               |
> | ---------- | ----------------------------------------------------------------------- |
> | 128      | latent map channels (z\_0) used as the diffusion target               |
> | 32       | stochastic posterior vector dimension used for KL regularization      |
> | 512      | projected decoder / conditioning working width after input projection |
>
> We will make this distinction explicit and use the notation consistently throughout the paper.
>
> We thank the reviewer again. These comments helped us sharpen the paper’s soundness and reproducibility claims.

---

> > ### Author Rebuttal · Reviewer_m831 · 2026-04-03
> >
> > Thank you for the response. I have nothing further to add.

---

> > > ### Author Response · Authors · 2026-04-03
> > >
> > > Dear Reviewer m831,
> > >
> > > Thank you very much for your time, follow-up, and thoughtful engagement with our rebuttal. We are glad that our clarifications helped address your concerns.
> > >
> > > We will carefully incorporate all the promised revisions into the final version, especially the clarified objective formulation, the corrected preprocessing protocol, and the zero-shot evaluation details to further improve soundness and reproducibility.
> > >
> > > Thank you again for your effort and support.
> > >
> > > Regards,
> > > Authors

---

### Decision · Program_Chairs · 2026-04-30

**Decision:**

Accept (regular)

**Comment:**

This paper proposes a temporary aware VAE-diffusion model for the cross model synthesis of phonocardiograms from ECG. Initial reviews rised valid concerns regarding the need for stronger baseline comparisons against diffusion Transformers and modern time series models, clarity on the ELBO derivation, and the framing of the models clinical utility. The authors however provided a convincing rebuttal, that included the requested baselines while appropriately narrowing their clients regarding diagnostic readiness. Since all the reviewers pound their concerns adequately addressed and convert on favourable scores, I agree that this is a well evaluated contribution to bio signal generation and therefore recommend acceptance.